# Probabilistic Graph Rewiring via Virtual Nodes

**Chendi Qian**[1*]   **Andrei Manolache**[234*]   **Christopher Morris**[1†]   **Mathias Niepert**[23†]

[1]Computer Science Department, RWTH Aachen University, Germany
[2]Computer Science Department, University of Stuttgart, Germany
[3]IMPRS-IS   [4]Bitdefender, Romania
chendi.qian@log.rwth-aachen.de
andrei.manolache@ki.uni-stuttgart.de

## Abstract

Message-passing graph neural networks (MPNNs) have emerged as a powerful paradigm for graph-based machine learning. Despite their effectiveness, MPNNs face challenges such as under-reaching and over-squashing, where limited receptive fields and structural bottlenecks hinder information flow in the graph. While graph transformers hold promise in addressing these issues, their scalability is limited due to quadratic complexity regarding the number of nodes, rendering them impractical for larger graphs. Here, we propose *implicitly rewired message-passing neural networks* (IPR-MPNNs), a novel approach that integrates *implicit* probabilistic graph rewiring into MPNNs. By introducing a small number of virtual nodes, i.e., adding additional nodes to a given graph and connecting them to existing nodes, in a differentiable, end-to-end manner, IPR-MPNNs enable long-distance message propagation, circumventing quadratic complexity. Theoretically, we demonstrate that IPR-MPNNs surpass the expressiveness of traditional MPNNs. Empirically, we validate our approach by showcasing its ability to mitigate under-reaching and over-squashing effects, achieving state-of-the-art performance across multiple graph datasets. Notably, IPR-MPNNs outperform graph transformers while maintaining significantly faster computational efficiency.

## 1 Introduction

*Message-passing graph neural networks* (MPNNs) [Gilmer et al., 2017, Scarselli et al., 2008] recently emerged as the most prominent machine-learning architecture for graph-structured, applicable to a large set of domains where data is naturally represented as graphs, such as bioinformatics [Jumper et al., 2021, Wong et al., 2023], social network analysis [Easley et al., 2012], and combinatorial optimization [Cappart et al., 2023, Qian et al., 2024].

MPNNs have been studied extensively in theory and practice [Böker et al., 2023, Gilmer et al., 2017, Kipf and Welling, 2017, Maron et al., 2019b, Morris et al., 2019, 2021, 2023, Veličković et al., 2018, Xu et al., 2019]. Recent works have shown that MPNNs suffer from over-squashing [Alon and Yahav, 2021], where bottlenecks arise from stacking multiple layers leading to large receptive fields, and under-reaching [Barceló et al., 2020], where distant nodes fail to communicate effectively because MPNNs' receptive fields are too narrow. These phenomena become prevalent when dealing with graphs with a large diameter, potentially hindering the performance of MPNNs on essential applications that depend on long-range interactions, such as protein folding [Gromiha and Selvaraj, 1999]. However, modeling long-range interactions in atomistic systems such as proteins remains a

---

*These authors contributed equally.
†Co-senior authorship.

Figure 1: Overview of how IPR-MPNNs implicitly rewire a graph through adding virtual nodes. IPR-MPNNs use an *upstream MPNN* to learn priors $\boldsymbol{\theta}$ for connecting original nodes with virtual nodes via edges, parameterizing a probability mass function conditioned on exactly-$k$ constraints. Subsequently, we sample exactly $k$ edges from this distribution for each original node, connecting it to $k$ virtual nodes. We input the resulting graph to a *downstream model*, typically an MPNN, for the final predictions task, propagating information from (1) original nodes to virtual nodes, (2) among virtual nodes, and (3) among original nodes. On the backward pass, the gradients of the loss $\ell$ regarding the parameters $\boldsymbol{\theta}$ are approximated through the derivative of the exactly-$k$ marginals.

challenging problem often solved in an ad-hoc fashion using coarse-graining methods [Saunders and Voth, 2013, Husic et al., 2020], effectively grouping the input nodes into cluster nodes.

Recently, *graph rewiring* [Bober et al., 2022, Deac et al., 2022, Gutteridge et al., 2023, Karhadkar et al., 2022, Shirzad et al., 2023, Topping et al., 2021] techniques emerged, adapting the graph structure to enhance connectivity and reduce node distance through methods ranging from edge additions to leveraging spectral properties and expander graphs. However, these approaches typically employ heuristic methods for selecting node pairs to rewire. Furthermore, *graph transformers* (GTs) [Chen et al., 2022a, Dwivedi et al., 2022b, He et al., 2023, Müller et al., 2023, Müller and Morris, 2024, Rampášek et al., 2022] and adaptive techniques like those by Errica et al. [2023] improve handling of long-range relationships but face challenges of quadratic complexity and extensive parameter sets, limiting their scalability.

Most similar to IPR-MPNNs is the work of Qian et al. [2023], which, like IPR-MPNNs, leverage recent techniques in differentiable $k$-subset sampling [Ahmed et al., 2023] to learn to add or remove edges of a given graph. However, like GTs, their approach suffers from quadratic complexity due to their need to compute a score for each node pair.

**Present Work** Our proposed IPR-MPNN architecture advances end-to-end probabilistic adaptive graph rewiring. Unlike the PR-MPNN framework of Qian et al. [2023], which suffers from quadratic complexity since the edge distribution is modeled *explicitly*, IPR-MPNNs *implicitly* transmits information across different parts of a graph by learning to connect the existing graph with newly-added virtual nodes, effectively circumventing quadratic complexity; see Figure 1 for a high-level overview of IPR-MPNNs. Our contributions are as follows.

1. We introduce IPR-MPNNs, adding virtual nodes to graphs, and learn to rewire them to the existing nodes end-to-end. IPR-MPNNs successfully overcome the quadratic complexity of graph transformers and previous graph rewiring techniques.

2. Theoretically, we demonstrate that IPR-MPNNs exceed the expressive capacity of standard MPNNs, typically limited by the 1-dimensional Weisfeiler-–Leman algorithm.

3. Empirically, we show that IPR-MPNNs outperform standard MPNN and GT architectures on a large set of established benchmark datasets, all while maintaining significantly faster computational efficiency.

4. We show that IPR-MPNNs are reducing the total effective resistance [Black et al., 2023] of multiple molecular datasets while also significantly improving the layer-wise sensitivity

[Di Giovanni et al., 2023, Xu et al., 2018] between distant nodes when compared to the base model.

*In summary, IPR-MPNNs represent a significant advancement towards scalable and adaptable MPNNs. They enhance expressiveness and adaptability to various data distributions while scaling to large graphs effectively.*

## 1.1 Related Work

In the following, we discuss relevant related work.

**MPNNs**   Recently, MPNNs [Gilmer et al., 2017, Scarselli et al., 2008] emerged as the most prominent graph machine learning architecture. Notable instances of this architecture include, e.g., Duvenaud et al. [2015], Hamilton et al. [2017], and Veličković et al. [2018], which can be subsumed under the message-passing framework introduced in Gilmer et al. [2017]. In parallel, approaches based on spectral information were introduced in, e.g., Bruna et al. [2014], Defferrard et al. [2016], Gama et al. [2019], Kipf and Welling [2017], Levie et al. [2019], and Monti et al. [2017]—all of which descend from early work in Baskin et al. [1997], Goller and Küchler [1996], Kireev [1995], Merkwirth and Lengauer [2005], Micheli and Sestito [2005], Micheli [2009], Scarselli et al. [2008], and Sperduti and Starita [1997].

**Limitations of MPNNs**   MPNNs are inherently biased towards encoding local structures, limiting their expressive power [Morris et al., 2019, 2021, Xu et al., 2019]. Specifically, they are at most as powerful as distinguishing non-isomorphic graphs or nodes with different structural roles as the 1-*dimensional Weisfeiler–Leman algorithm* [Weisfeiler and Leman, 1968], a well-studied heuristic for the graph isomorphism problem; see Section C. Additionally, they cannot capture global or long-range information, often linked to phenomena such as under-reaching [Barceló et al., 2020] or over-squashing [Alon and Yahav, 2021], with the latter being heavily investigated in recent works.

**Graph Transformers**   Different from the above, graph transformers [Chen et al., 2022a,b, Dwivedi et al., 2022b, He et al., 2023, Hussain et al., 2022, Kim et al., 2022, Ma et al., 2023, Mialon et al., 2021, Müller et al., 2023, Müller and Morris, 2024, Rampášek et al., 2022, Shirzad et al., 2023] and similar global attention mechanisms [Liu et al., 2021, Wu et al., 2021] marked a shift from local to global message passing, aggregating over all nodes. While not understood in a principled way, empirical studies indicate that graph transformers possibly alleviate over-squashing; see Müller et al. [2023]. However, all transformers suffer from their quadratic space and memory requirements due to computing an attention matrix.

**Rewiring Approaches for MPNNs**   Several recent works aim to circumvent over-squashing via graph rewiring. The most straightforward way of graph rewiring is incorporating multi-hop neighbors. For example, Brüel-Gabrielsson et al. [2022] rewires the graphs with $k$-hop neighbors and virtual nodes and augments them with positional encodings. MixHop [Abu-El-Haija et al., 2019], SIGN [Frasca et al., 2020], DIGL [Gasteiger et al., 2019], and SP-MPNN [Abboud et al., 2022] can also be considered as graph rewiring as they can reach further-away neighbors in a single layer. Particularly, Gutteridge et al. [2023] rewires the graph similarly to Abboud et al. [2022] but with a novel delay mechanism, showcasing promising empirical results. Several rewiring methods depend on particular metrics, e.g., Ricci or Forman curvature [Bober et al., 2022] and balanced Forman curvature [Topping et al., 2021]. In addition, Deac et al. [2022], Shirzad et al. [2023] utilize expander graphs to enhance message passing and connectivity, while Karhadkar et al. [2022] resort to spectral techniques, and Banerjee et al. [2022] propose a greedy random edge flip approach to overcome over-squashing. DiffWire [Arnaiz-Rodríguez et al., 2022] conducts fully differentiable and parameter-free graph rewiring by leveraging the Lovász bound and spectral gap. Refining Topping et al. [2021], Di Giovanni et al. [2023] analyzed how the architectures' width and graph structure contribute to the over-squashing problem, showing that over-squashing happens among nodes with high commute time, stressing the importance of rewiring techniques. Contrary to our proposed method, these strategies to mitigate over-squashing rely on heuristic rewiring methods or purely randomized approaches that may not adapt well to a given prediction task. LASER [Barbero et al., 2023] performs graph rewiring while respecting the original graph structure. The recent work S2GCN [Geisler et al., 2024] combines spectral and spatial graph filters and implicitly introduces graph rewiring for message passing. Most

similar to IPR-MPNNs is the work of Qian et al. [2023], which, like IPR-MPNNs, leverage recent techniques in differentiable $k$-subset sampling [Ahmed et al., 2023] to learn to add or remove edges of a given graph. However, like GTs, their approach suffers from quadratic complexity due to their need to compute a score for each node pair. In addition to the differentiable $k$-subset sampling [Ahmed et al., 2023] method we use in this work, there are other gradient estimation approaches such as GUMBEL SOFTSUB-ST [Xie and Ermon, 2019] and I-MLE [Niepert et al., 2021, Minervini et al., 2023].

There is also a large set of works from graph structure learning proposing heuristical graph rewiring approaches and hierarchical MPNNs; see Section A for details.

## 2   Background

In the following, we introduce notation and formally define MPNNs.

**Notations**   Let $\mathbb{N} := \{1, 2, 3, \dots\}$. For $n \geq 1$, let $[n] := \{1, \dots, n\} \subset \mathbb{N}$. We use $\{\!\!\{\dots\}\!\!\}$ to denote multisets, i.e., the generalization of sets allowing for multiple instances for each of its elements. A *graph* $G$ is a pair $(V(G), E(G))$ with *finite* sets of *nodes* or *vertices* $V(G)$ and *edges* $E(G) \subseteq \{\!\{u, v\} \subseteq V(G) \mid u \neq v\}$. If not otherwise stated, we set $n := |V(G)|$, and the graph is of *order* $n$. We also call the graph $G$ an $n$-order graph. For ease of notation, we denote the edge $\{u, v\}$ in $E(G)$ by $(u, v)$ or $(v, u)$. Throughout the paper, we use standard notations, e.g., we denote the *neighborhood* of a vertex $v$ by $N(v)$ and $\ell(v)$ denotes its discrete vertex label, and so on; see Section B for details.

**Message-passing Graph Neural Networks**   Intuitively, MPNNs learn a vectorial representation, i.e., a $d$-dimensional real-valued vector, representing each vertex in a graph by aggregating information from neighboring vertices. Let $\boldsymbol{G} = (G, \boldsymbol{X})$ be an $n$-order attributed graph with node feature matrix $\boldsymbol{X} \in \mathbb{R}^{n \times d}$, for $d > 0$, following, Gilmer et al. [2017] and Scarselli et al. [2008], in each layer, $t > 0$, we compute vertex features

$$\boldsymbol{h}_v^{(t)} := \mathsf{UPD}^{(t)}\Big(\boldsymbol{h}_v^{(t-1)}, \mathsf{AGG}^{(t)}\big(\{\!\!\{\boldsymbol{h}_u^{(t-1)} \mid u \in N(v)\}\!\!\}\big)\Big) \in \mathbb{R}^d,$$

where $\mathsf{UPD}^{(t)}$ and $\mathsf{AGG}^{(t)}$ may be parameterized functions, e.g., neural networks, and $\boldsymbol{h}_v^{(t)} := \boldsymbol{X}_v$. In the case of graph-level tasks, e.g., graph classification, one uses

$$\boldsymbol{h}_G := \mathsf{READOUT}\big(\{\!\!\{\boldsymbol{h}_v^{(T)} \mid v \in V(G)\}\!\!\}\big) \in \mathbb{R}^d,$$

to compute a single vectorial representation based on learned vertex features after iteration $T$. Again, READOUT may be a parameterized function, e.g., a neural network. To adapt the parameters of the above three functions, they are optimized end-to-end, usually through a variant of stochastic gradient descent, e.g., Kingma and Ba [2015], together with the parameters of a neural network used for classification or regression.

## 3   Implicit Probabilistically Rewired MPNNs

*Implicit probabilistically rewired message-passing neural networks* (IPR-MPNNs) learn a probability distribution over edges connecting the *original nodes* of a graph to additional *virtual nodes*, providing an *implicit* way of enhancing the graph connectivity. To learn to rewire original nodes with these added virtual nodes, IPR-MPNNs use an *upstream model*, usually an MPNN, to generate scores or (unnormalized) priors $\boldsymbol{\theta} := h_{\boldsymbol{u}}(\boldsymbol{A}(G), \boldsymbol{X}) \in \mathbb{R}^{n \times m}$, where $G$ is an $n$-order graph with adjacency matrix $\boldsymbol{A}(G) \in \{0, 1\}^{n \times n}$, node feature matrix $\boldsymbol{X} \in \mathbb{R}^{n \times d}$, for $d > 0$, and number of virtual nodes $m$ with $m \ll n$.

IPR-MPNNs use the priors $\boldsymbol{\theta}$ from the upstream model to sample new edges between the original nodes and the $m$ virtual nodes from the posterior constrained to exactly $k$ edges, thus obtaining an *assignment matrix* $\boldsymbol{H} \in \{0, 1\}^{n \times m}$, i.e., an adjacency matrix between the input and virtual nodes. The assignment matrix $\boldsymbol{H}$ is then used in a *downstream model* $f_{\boldsymbol{v}}$, utilized for solving our downstream task, e.g., graph-level classification or regression. Leveraging recent advancements in gradient estimation for $k$-subset sampling [Ahmed et al., 2023], the upstream and downstream models are jointly optimized, enabling the model to be trained end-to-end; see below.

Hence, unlike PR-MPNNs [Qian et al., 2023], which in the worst case *explicitly* model an edge distribution for all $n^2$ possible edge candidates for rewiring, IPR-MPNNs leverage virtual nodes for implicit rewiring and passing long-range information. Therefore, IPR-MPNNs benefit from better computation complexity while being more expressive than the 1-WL; see Section 4. Moreover, intuitively, it is also easier to model a distribution of edges connected to a few virtual nodes than to learn the explicit distribution of all possible edges in a graph. More specifically, while in PR-MPNNs, the priors $\boldsymbol{\theta}$ can have a size of up to $n^2$ for an $n$-order graph, IPR-MPNNs use $m \cdot n$ parameters, therefore significantly enhancing both computational efficiency and model simplicity.

In the following, we describe the IPR-MPNN architecture in detail.

**Sampling Edges** Let $\mathfrak{A}_n$ represent the adjacency matrices of $n$-order graphs. Consider $(G, \boldsymbol{X})$ as a graph of order $n$ with adjacency matrix $\boldsymbol{A}(G) \in \mathfrak{A}_n$ and node feature matrix $\boldsymbol{X} \in \mathbb{R}^{n \times d}$, for $d > 0$. IPR-MPNNs maintain a parameterized upstream model $h_{\boldsymbol{u}} \colon \mathfrak{A}_n \times \mathbb{R}^{n \times d} \to \Theta$, usually implemented through an MPNN and parameterized by $\boldsymbol{u}$. The upstream model transforms an adjacency matrix along with its node attributes into a set of unnormalized node priors $\boldsymbol{\theta} \in \Theta \subseteq \mathbb{R}^{n \times m}$, with $m$ denoting the predefined number of virtual nodes. Formally,

$$\boldsymbol{\theta} \coloneqq h_{\boldsymbol{u}}(\boldsymbol{A}(G), \boldsymbol{X}) \in \mathbb{R}^{n \times m}.$$

The matrix of priors $\boldsymbol{\theta}$ serves as the parameter matrix for the conditional probability mass function from which the assignment matrix $\boldsymbol{H} \in \{0, 1\}^{n \times m}$ is sampled. Crucially and contrary to prior work [Qian et al., 2023], these edges connect the input and a *small* number $m$ of virtual nodes. Hence, each row of matrix $\boldsymbol{\theta}_{i:} \in \mathbb{R}^m$ represents the unnormalized probability of assigning node $i$ to each virtual node. Formally, we have,

$$p_{\boldsymbol{\theta}}(\boldsymbol{H}_{i:}) \coloneqq \prod_{j=1}^{M} p_{\boldsymbol{\theta}_{ij}}(\boldsymbol{H}_{ij}), \text{ for } i \in [n],$$

where $p_{\boldsymbol{\theta}_{ij}}(\boldsymbol{H}_{ij} = 1) = \text{sigmoid}(\boldsymbol{\theta}_{ij})$ and $p_{\boldsymbol{\theta}_{ij}}(\boldsymbol{H}_{ij} = 0) = 1 - \text{sigmoid}(\boldsymbol{\theta}_{ij})$. Without loss of generality, we allow each node to be assigned to $k$ virtual nodes, with $k \in [m]$. That is, each row of the sampled assignment matrix has exactly $k$ non-zero entries, i.e.,

$$p_{(\boldsymbol{\theta}, k)}(\boldsymbol{H}) \coloneqq \begin{cases} p_{\boldsymbol{\theta}}(\boldsymbol{H})/Z & \text{if } \|\boldsymbol{H}_{i:}\|_1 = k, \text{for all } i \in [n], \\ 0 & \text{otherwise,} \end{cases} \quad \text{with} \quad Z \coloneqq \sum_{\substack{\boldsymbol{B} \in \{0,1\}^{n \times m} : \\ \|\boldsymbol{B}_{i:}\|_1 = k, \forall\, i \in [n]}} p_{\boldsymbol{\theta}}(\boldsymbol{B}).$$

We can potentially sample independently $q$ times $\boldsymbol{H}^{(i)} \sim p_{(\boldsymbol{\theta}, k)}(\boldsymbol{H})$ and consequently obtain $q$ multiple assignment matrices $\bar{\boldsymbol{H}} \coloneqq \{\!\!\{ \boldsymbol{H}^{(1)}, \boldsymbol{H}^{(2)}, \ldots, \boldsymbol{H}^{(q)} \}\!\!\}$, which, together with corresponding number of copies of $\boldsymbol{A}(G)$ and $\boldsymbol{X}$, will be utilized by the downstream model for the tasks.

**Message-passing Architecture of IPR-MPNNs** Here, we outline the message-passing scheme after adding virtual nodes and edges. Consider an $n$-order graph $G$ and a virtual node set $C(G)$ of cardinality $m$, where each original node $v \in V(G) \coloneqq [n]$ is assigned to $k \in [m]$ virtual nodes. We assign original nodes $v$ to a subset of the virtual node using the function $a \colon V(G) \to [C(G)]_k$, where $v \mapsto \{c \in C(G) \mid \boldsymbol{H}_{vc} = 1\}$ and $[C(G)]_k$ is the set of $k$-element subsets of the virtual nodes. Conversely, each virtual node $c \in C(G)$ links to several original nodes. Hence, we define an inverse assignment as the set of all original nodes assigned to virtual node $c$, i.e., $a^{-1}(c) \coloneqq \{v \in V(G) \mid c \in a(v)\}$. Across the graph, the union of these inverse assignments equals the set of original nodes, i.e., $\bigcup_c a^{-1}(c) = V(G)$.

We represent the embedding of an original node $v \in V(G)$ at any given layer $t \geq 0$ as $\boldsymbol{h}_v^{(t)}$, and similarly, the embedding for a virtual node $c \in C(G)$ as $\boldsymbol{g}_c^{(t)}$. To compute these embeddings, IPR-MPNNs compute initial embeddings for each virtual node. Subsequently, the architecture updates the virtual nodes via the adjacent original nodes' embeddings. Afterward, the virtual nodes exchange messages, and finally, the virtual nodes update adjacent original nodes' embeddings. Below, we outline the steps in order of execution involved in our message-passing algorithm in detail.

**Intializing Virtual Node Embeddings**    Before executing inter-hierarchical message passing, we need to initialize the embeddings of virtual nodes. To that, given the node assignments $a(v)$, for $v \in V(G)$, we can effectively divide the original graph into several subgraphs, where nodes sharing the same assignment label are grouped together to form an induced subgraph. Formally, for any virtual node $c \in C(G)$, we have the induced subgraph $G_c$ with node subset $V_c(G_c) := \{v \mid v \in V(G) \cap a^{-1}(c)\}$ and edge set $E_c(G_c) := \{\{u, v\} \mid \{u, v\} \in E(G), u \in a^{-1}(c) \text{ and } v \in a^{-1}(c)\}$. The initial attributes of the virtual node $c$ are defined by the node features of its corresponding subgraph $G_c$, calculated using an MPNN, i.e.,

$$\boldsymbol{g}_c^{(0)} := \mathsf{MPNN}\,(G_c), \text{ for } c \in C(G). \tag{1}$$

Alternatively, we can generate random features for each virtual node as initial node features or assign unique identifiers to them.

**Updating Virtual Nodes**    In each step $t > 0$, we collect the embeddings from original nodes to virtual nodes according to their assignments and obtain intermediate virtual node embeddings

$$\bar{\boldsymbol{g}}_c^{(t)} := \mathsf{AGGn}^{(t)}\left(\{\!\!\{\boldsymbol{h}_v^{(t-1)} \mid v \in a^{-1}(c)\}\!\!\}\right), \text{ for } c \in C(G),$$

where $\mathsf{AGGn}^{(t)}$ denotes some permutation-equivariant aggregation function designed for multisets.

**Updating Among Virtual Nodes**    We assume the virtual nodes form a complete, undirected, unweighted graph, and we perform message passing among the virtual nodes to update their embeddings. That is, at step $t$, we set

$$\boldsymbol{g}_c^{(t)} := \mathsf{UPDc}^{(t)}\left(\boldsymbol{g}_c^{(t-1)}, \bar{\boldsymbol{g}}_c^{(t)}, \mathsf{AGGc}^{(t)}\left(\{\!\!\{\bar{\boldsymbol{g}}_j^{(t)} \mid j \in C(G), j \neq c\}\!\!\}\right)\right),$$

where $\mathsf{UPDc}$ and $\mathsf{AGGc}$ are the update and neighborhood aggregation functions for virtual nodes.

**Updating Original Nodes**    Finally, we redistribute the embeddings from the virtual nodes back to the base nodes. This update process considers both the neighbors in the original graph and the virtual nodes to which the original nodes are assigned. The updating mechanism is detailed in the following equation,

$$\boldsymbol{h}_v^{(t)} := \mathsf{UPD}^{(t)}\left(\boldsymbol{h}_v^{(t-1)}, \mathsf{AGG}^{(t)}\left(\{\!\!\{\boldsymbol{h}_u^{(t-1)} \mid u \in N(v)\}\!\!\}\right), \mathsf{DS}^{(t)}\left(\{\!\!\{\boldsymbol{g}_c^{(t)} \mid c \in a(v)\}\!\!\}\right)\right), \tag{2}$$

where $\mathsf{UPD}$ is the update function, $\mathsf{AGG}$ is the aggregation function, and $\mathsf{DS}$ is the distributing function that incorporates embeddings from virtual nodes back to the original nodes.

**Gradient Estimation**    In the context of our downstream MPNN model described above, we define the set of its learnable parameters as $\boldsymbol{v}$ and group these with the upstream model parameters $\boldsymbol{u}$ into a combined tuple $\boldsymbol{w} = (\boldsymbol{u}, \boldsymbol{v})$. We express the downstream model as $f_{\boldsymbol{v}}$, resulting in the loss function

$$L\left(\boldsymbol{A}(G), \boldsymbol{X}, \bar{\boldsymbol{H}}; \boldsymbol{w}\right) := \mathbb{E}_{\boldsymbol{H}^{(i)} \sim p_{(\boldsymbol{\theta}, k)}(\boldsymbol{H})}\left[\ell\left(f_{\boldsymbol{v}}\left(\boldsymbol{A}(G), \boldsymbol{X}, \{\!\!\{\boldsymbol{H}^{(1)}, \dots, \boldsymbol{H}^{(q)}\}\!\!\}\right), y\right)\right].$$

While the gradients for the downstream model $f_{\boldsymbol{v}}$ can be straightforwardly calculated via backpropagation, obtaining gradients for the upstream model parameters $\boldsymbol{\theta}$ is more challenging, as the assignment matrices $\bar{\boldsymbol{H}}$ are sampled from the priors $\boldsymbol{\theta}$, a process which is not differentiable.

Similar to prior work [Qian et al., 2023], we utilize SIMPLE [Ahmed et al., 2023], which efficiently estimates gradients under $k$-subset constraints. This method involves exact sampling in the forward phase and uses the marginal of the priors $\mu(\boldsymbol{\theta}) \in \mathbb{R}^{n \times m}$ during the backward phase to approximate gradients $\nabla_{\boldsymbol{\theta}} L \approx \partial_{\boldsymbol{\theta}} \mu(\boldsymbol{\theta}) \nabla_{\boldsymbol{H}} \ell$.

The following analysis shows that our proposed method circumvents the problem of being quadratic in the number of input nodes.

**Complexity**    Assuming a constant number of hidden dimensions and layers of the MPNNs, recall that the runtime complexity of a plain MPNN is $\mathcal{O}(|E|)$, where $|E|$ is the number of edges of a given graph. In the IPR-MPNN framework, we still have an MPNN backbone, augmented with

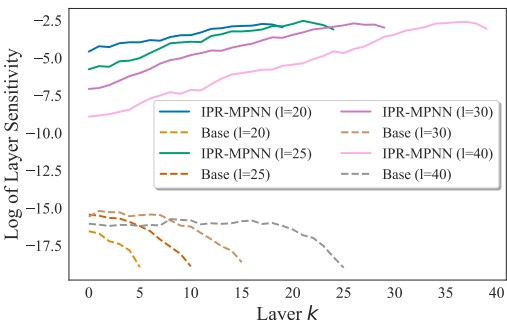 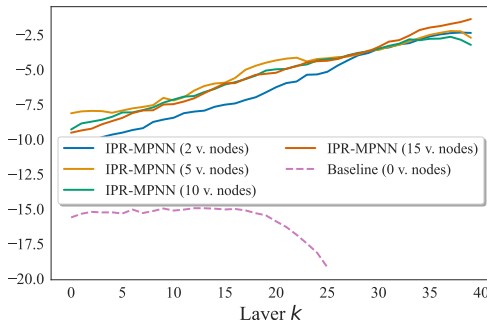

Figure 2: Comparing model sensitivity across different layers for the two most distant nodes from graphs from the ZINC dataset. On the left, we compare the sensitivity for models with a varying number of layers. We can observe that IPR-MPNNs maintain a high sensitivity even for the last layer, while the base models have the sensitivity decaying to 0. On the right, we compare models with a different number of virtual nodes, observing that the results are similar for all of the variants.

inter-message passing involving virtual nodes. Hence, obtaining priors for the original nodes via an MPNN or even a simple MLP has a complexity of $\mathcal{O}(|E| + n \cdot m)$, where $m$ is the number of candidate virtual nodes to be selected. Sampling the node assignment with Ahmed et al. [2023] is in $\mathcal{O}(n \cdot \log m \cdot \log k)$. The message aggregation and distribution between base nodes and virtual nodes have complexity $\mathcal{O}(n \cdot k)$, where $k \in [m]$ is the number of nodes a node is assigned to. Finally, the intra-virtual node message passing is in $\mathcal{O}(m^2)$, as they are fully connected. In summary, IPR-MPNNs have a running time complexity in $\mathcal{O}(|E| + n \cdot m + m^2)$. Since $m \ll n$ is a small constant, IPR-MPNNs show great potential due to their low complexity compared to the quadratic worst-case complexities of graph transformers [Müller et al., 2023] and other rewiring methods, e.g., Gutteridge et al. [2023], Qian et al. [2023].

## 4   Expressive Power

In this section, we analyze the extent to which IPR-MPNNs can separate non-isomorphic graphs on which the 1-WL isomorphism test fails [Xu et al., 2019, Morris et al., 2019], and whether IPR-MPNNs can maintain isomorphisms between pairs of graphs. We adopt the notion of probabilistic separation from Qian et al. [2023].

Our arguments rely on the ability of our upstream MPNN to assign arbitrary and distinct exactly-$k$ distributions for each color class. By modifying the graph structure, we can make the rewired graphs 1-WL-distinguishable, enabling our downstream MPNN to separate them. However, since the expressiveness of the upstream model is also equivalent to 1-WL, there is a possibility of still separating isomorphic graphs.

The following result demonstrates that we can preserve almost all partial subgraph isomorphisms.

**Theorem 4.1.** *Let $k > 0$, $\varepsilon \in (0, 1)$, and $G$, $H$ be two graphs with identical 1-WL stable colorings. Let $M$ be the set of ordered virtual nodes, $V_G$ and $V_H$ be the subset of nodes in $G$ and $H$ that have a color class of cardinality 1, with $|V_G| = |V_H| = d$, and $W_G$, $W_H$ the subset of nodes that have a color class of cardinality greater than 1, with $|W_G| = |W_H| = n$. Then, for all choices of 1-WL-equivalent functions $f$,*

*(1) there exists a conditional probability mass function $p_{(\boldsymbol{\theta}, k)}$ that does* not *separate $G[V_G]$ and $H[V_H]$ with probability at least $1 - \varepsilon$.*

*(2) There exists a conditional probability mass function $p_{(\boldsymbol{\theta}, k)}$ that separates $G[W_G]$ and $H[W_H]$ with probability strictly greater than $0$.*

We argue that preserving these partial subgraph isomorphisms is sufficient for most examples in practice. Indeed, our empirical findings show that we can successfully solve both the EXP and CSL datasets, whereas a 1-WL model obtains random performance; see Table A11, Table A10.

The next corollary follows the above and recovers Theorem 4.1 from Qian et al. [2023]. The corollary tells us that, even if there are isomorphic graphs that we risk making separable, we will maintain the isomorphism between almost all isomorphic pairs.

Table 1: We compare IPR-MPNN on `QM9` with the base downstream GIN model [Xu et al., 2019], two graph rewiring techniques [Gutteridge et al., 2023, Qian et al., 2023], a multi-hop MPNN [Abboud et al., 2022], and the relational GIN [Schlichtkrull et al., 2018]. The best-performing method is colored in green, the second-best in blue, and third orange. IPR-MPNN obtains the best result on all targets, except for MU, where it obtains the second-best result.

| PROPERTY | GIN [2019] | R-GIN+FA [2018] | SPN [2022] | DREW-GIN [2023] | PR-MPNN [2023] | IPR-MPNN |
|---|---|---|---|---|---|---|
| MU | $2.64_{\pm 0.01}$ | $2.54_{\pm 0.09}$ | $2.32_{\pm 0.28}$ | $1.93_{\pm 0.06}$ | $1.99_{\pm 0.02}$ | $2.01_{\pm 0.01}$ |
| ALPHA | $7.67_{\pm 0.16}$ | $2.28_{\pm 0.04}$ | $1.77_{\pm 0.09}$ | $1.63_{\pm 0.03}$ | $2.28_{\pm 0.06}$ | $1.36_{\pm 0.01}$ |
| HOMO | $1.70_{\pm 0.02}$ | $1.26_{\pm 0.02}$ | $1.26_{\pm 0.09}$ | $1.16_{\pm 0.01}$ | $1.14_{\pm 0.01}$ | $1.07_{\pm 0.03}$ |
| LUMO | $3.05_{\pm 0.01}$ | $1.34_{\pm 0.04}$ | $1.19_{\pm 0.05}$ | $1.13_{\pm 0.02}$ | $1.12_{\pm 0.01}$ | $1.03_{\pm 0.07}$ |
| GAP | $3.37_{\pm 0.03}$ | $1.96_{\pm 0.04}$ | $1.89_{\pm 0.11}$ | $1.74_{\pm 0.02}$ | $1.70_{\pm 0.01}$ | $1.61_{\pm 0.08}$ |
| R2 | $23.35_{\pm 1.08}$ | $12.61_{\pm 0.37}$ | $10.66_{\pm 0.40}$ | $9.39_{\pm 0.13}$ | $10.41_{\pm 0.35}$ | $8.17_{\pm 0.53}$ |
| ZPVE | $66.87_{\pm 1.45}$ | $5.03_{\pm 0.36}$ | $2.77_{\pm 0.17}$ | $2.73_{\pm 0.19}$ | $4.73_{\pm 0.08}$ | $1.96_{\pm 0.07}$ |
| U0 | $21.48_{\pm 0.17}$ | $2.21_{\pm 0.12}$ | $1.12_{\pm 0.13}$ | $1.01_{\pm 0.09}$ | $2.23_{\pm 0.13}$ | $0.74_{\pm 0.11}$ |
| U | $21.59_{\pm 0.30}$ | $2.32_{\pm 0.18}$ | $1.03_{\pm 0.09}$ | $0.99_{\pm 0.08}$ | $2.31_{\pm 0.06}$ | $0.79_{\pm 0.12}$ |
| H | $21.96_{\pm 1.24}$ | $2.26_{\pm 0.19}$ | $1.05_{\pm 0.04}$ | $1.06_{\pm 0.09}$ | $2.66_{\pm 0.01}$ | $0.75_{\pm 0.14}$ |
| G | $19.53_{\pm 0.47}$ | $2.04_{\pm 0.24}$ | $0.97_{\pm 0.06}$ | $1.06_{\pm 0.14}$ | $2.24_{\pm 0.01}$ | $0.62_{\pm 0.13}$ |
| Cv | $7.34_{\pm 0.06}$ | $1.86_{\pm 0.03}$ | $1.36_{\pm 0.06}$ | $1.24_{\pm 0.02}$ | $1.44_{\pm 0.01}$ | $1.03_{\pm 0.04}$ |
| OMEGA | $0.60_{\pm 0.03}$ | $0.80_{\pm 0.04}$ | $0.57_{\pm 0.04}$ | $0.55_{\pm 0.01}$ | $0.48_{\pm 0.00}$ | $0.45_{\pm 0.03}$ |

Table 2: Results on the PEPTIDES and PCQM-CONTACT datasets from the long-range graph benchmark [Dwivedi et al., 2022b]. For PCQM-CONTACT, we use both the initially proposed evaluation methodology (PCQM$^{(1)}$) and the filtering methodologies proposed in Tönshoff et al. [2023] (PCQM$^{(2)}$ for filtering and PCQM$^{(3)}$ for extended filtering). Green is the best model, blue is the second, and red the third. IPR-MPNNs obtain the best predictive performance on all datasets.

| MODEL | PEPTIDES-FUNC ↑ | PEPTIDES-STRUCT ↓ | PCQM$^{(1)}$ ↑ | PCQM$^{(2)}$ ↑ | PCQM$^{(3)}$ ↑ |
|---|---|---|---|---|---|
| GINE [2019, 2023] | $0.6621_{\pm 0.0067}$ | $0.2473_{\pm 0.0017}$ | $0.3509_{\pm 0.0006}$ | $0.3725_{\pm 0.0006}$ | $0.4617_{\pm 0.0005}$ |
| GCN [2017, 2023] | $0.6860_{\pm 0.0050}$ | $0.2460_{\pm 0.0007}$ | $0.3424_{\pm 0.0007}$ | $0.3631_{\pm 0.0006}$ | $0.4526_{\pm 0.0006}$ |
| DREW [2023] | $0.7150_{\pm 0.0044}$ | $0.2536_{\pm 0.0015}$ | $0.3444_{\pm 0.0017}$ | - | - |
| PR-MPNN [2023] | $0.6825_{\pm 0.0086}$ | $0.2477_{\pm 0.0005}$ | - | - | - |
| AMP [2023] | $0.7163_{\pm 0.0058}$ | $0.2431_{\pm 0.0004}$ | - | - | - |
| NBA [2023] | $0.7207_{\pm 0.0028}$ | $0.2472_{\pm 0.0008}$ | - | - | - |
| GATEDGCN+PE+VN$_G$ [2024] | $0.6822_{\pm 0.0052}$ | $0.2458_{\pm 0.0006}$ | - | - | - |
| S2GCN [2024] | $0.7275_{\pm 0.0066}$ | $0.2467_{\pm 0.0019}$ | - | - | - |
| S2GCN+PE [2024] | $0.7311_{\pm 0.0066}$ | $0.2447_{\pm 0.0007}$ | - | - | - |
| GPS [2022] | $0.6535_{\pm 0.0041}$ | $0.2509_{\pm 0.0014}$ | $0.3498_{\pm 0.0005}$ | $0.3722_{\pm 0.0005}$ | $0.4703_{\pm 0.0014}$ |
| EXPHORMER [2023] | $0.6527_{\pm 0.0043}$ | $0.2481_{\pm 0.0007}$ | $0.3637_{\pm 0.0020}$ | - | - |
| GRIT [2023] | $0.6988_{\pm 0.0082}$ | $0.2460_{\pm 0.0012}$ | - | - | - |
| GRAPH MLP-MIXER [2023] | $0.6970_{\pm 0.0080}$ | $0.2475_{\pm 0.0015}$ | - | - | - |
| GRAPH VIT [2023] | $0.6942_{\pm 0.0075}$ | $0.2449_{\pm 0.0016}$ | - | - | - |
| IPR-MPNN (OURS) | $0.7210_{\pm 0.0039}$ | $0.2422_{\pm 0.0007}$ | $0.3670_{\pm 0.0082}$ | $0.3846_{\pm 0.0047}$ | $0.4756_{\pm 0.0035}$ |

**Corollary 4.1.1.** *For sufficiently large $n$, for every $\varepsilon \in (0, 1)$, a set $m$ of ordered virtual nodes, and $k > 0$, we have that almost all pairs, in the sense of Babai et al. [1980], of isomorphic $n$-order graphs $G$ and $H$ and all permutation-invariant, 1-$\mathcal{WL}$-equivalent functions $f : \mathfrak{A}_n \to \mathbb{R}^d$, $d > 0$, there exists a probability mass function $p_{(\boldsymbol{\theta}, k)}$ that separates the graph $G$ and $H$ with probability at most $\varepsilon$ regarding $f$.*

The previous theorems show that we are preserving isomorphisms better than purely randomized approaches while being more powerful than 1-$\mathcal{WL}$ since we can separate non-isomorphic graphs with a probability strictly greater than 0. We provide the proofs and examples in Section E.

## 5 Experimental Setup and Results

To empirically validate the effectiveness of our IPR-MPNN framework, we conducted a series of experiments on both synthetic and real-world molecular datasets, answering the following research questions. An open repository of our code can be accessed at `https://github.com/chendiqian/IPR-MPNN`.

**Q1** Do IPR-MPNNs alleviate over-squashing and under-reaching?

**Q2** Do IPR-MPNNs demonstrate enhanced expressivity compared to MPNNs?

**Q3** How do IPR-MPNNs compare in predictive performance on molecular datasets against other rewiring methods and graph transformers?

**Q4** Does the lower theoretical complexity of IPR-MPNNs translate to faster runtimes in practice?

Table 3: IPR-MPNN training, inference (seconds per epoch), and memory consumption statistics in comparison to the base GINE model [Xu et al., 2019], the GPS graph transformer [Rampášek et al., 2022] and the Drew model [Gutteridge et al., 2023] on the PEPTIDES-STRUCT dataset [Dwivedi et al., 2022b]. Our model has almost the same computation and memory efficiency as the base GINE model while being twice as fast and significantly more memory efficient when compared to GPS.

|  | GINE | IPR-MPNN | GPS | DREW |
|---|---|---|---|---|
| # PAR. | $503k$ | $536k$ | $558k$ | $522k$ |
| TRN S/EP. | $2.68\pm0.01$ | $2.98\pm0.02$ | $7.81\pm0.32$ | $3.20\pm0.03$ |
| VAL S/EP. | $0.21\pm0.00$ | $0.27\pm0.00$ | $0.58\pm0.04$ | $0.36\pm0.00$ |
| MEM. | 1.7GB | 1.9GB | 22.2GB | 1.8GB |

**Datasets, Experimental Results, and Discussion**   To address **Q1**, we investigate whether our method alleviates over-squashing and under-reaching by experimenting on TREES-NEIGHBOURSMATCH [Alon and Yahav, 2021] and TREES-LEAFCOUNT [Qian et al., 2023]. On TREES-LEAFCOUNT with a tree depth of four, we obtain perfect performance on the test dataset with a one-layer downstream network, indicating we can alleviate under-reaching. Furthermore, on TREES-NEIGHBOURSMATCH, our method obtains perfect performance to a depth up to six, effectively alleviating over-squashing, as shown in Figure A4. To quantitatively assess whether over-squashing is mitigated in real-world scenarios, we computed the average layer-wise sensitivity [Xu et al., 2018, Di Giovanni et al., 2023, Errica et al., 2023] between the most distant nodes in graphs from the ZINC dataset and compared these results with those from the baseline GINE model. Specifically, we compute the logarithm of the symmetric sensitivity between the most distant nodes $u, v$ as $\log\left(|\partial \mathbf{h}_v^l/\partial \mathbf{h}_u^k + \partial \mathbf{h}_u^l/\partial \mathbf{h}_v^k|\right)$, where $k$ to $l$ represent the intermediate layers. We show that IPR-MPNNs maintain a high layer-wise sensitivity compared to the base model, as seen in Figure 2, implying that they can successfully account for long-range relationships, even with multiple stacked layers. Lastly, we measured the average total effective resistance [Black et al., 2023] of five molecular datasets before and after rewiring, showing in Figure 3 that IPR-MPNNs are successfully improving connectivity by reducing the average total effective resistance of all evaluated datasets.

For **Q2**, we conduct experiments on the EXP [Abboud et al., 2020] and CSL [Murphy et al., 2019] datasets to evaluate the expressiveness of IPR-MPNNs. The results, as detailed in Table A10 and Table A11, demonstrate that our IPR-MPNN framework handles these datasets effectively and exhibits improved expressiveness over the base 1-WL-equivalent GIN model.

For answering **Q3**, we utilize several real-world molecular datasets—QM9 [Hamilton et al., 2017], ZINK 12K [Jin et al., 2017], OGB-MOLHIV [Hu et al., 2020], TUDATASET [Morris et al., 2020a], and datasets from the LONG-RANGE GRAPH BENCHMARK [Dwivedi et al., 2022b], namely PEPTIDES and PCQM-CONTACT. Our results demonstrate that IPR-MPNNs effectively account for long-range relationships, achieving state-of-the-art performance on the PEPTIDES and PCQM-CONTACT datasets, as detailed in Table 2. Notably, on the PCQM-CONTACT link prediction tasks, IPR-MPNNs outperform all other candidates across three measurement metrics outlined in Tönshoff et al. [2023]. For QM9, we show in Table 1 that IPR-MPNNs greatly outperform similar methods, obtaining the best results on 12 of 13 target properties. On ZINC and OGB-MOLHIV, we outperform similar MPNNs and graph transformers, namely GPS Rampášek et al. [2022] and SAT [Chen et al., 2022a], obtaining state-of-the-art results; see Table 4. For the TUDATASET collection, we achieve the best results on four of the five molecular datasets; see Table A9.

Finally, to address **Q4**, we evaluate the computation time and memory usage of IPR-MPNNs in comparison with the GPS graph transformer [Rampášek et al., 2022] on PEPTIDES-STRUCT and extend our analysis to include PR-MPNNs [Qian et al., 2023], SAT [Chen et al., 2022a], and GPS on the ZINC dataset. The results in Tables 3 and A12 demonstrate that IPR-MPNNs adhere to their theoretical linear runtime complexity in practice. We observed a notable speedup in training and validation times per epoch while reducing the memory footprint by a large margin compared to the two mentioned transformers. This efficiency underscores the practical advantages of IPR-MPNNs in computational speed and resource utilization.

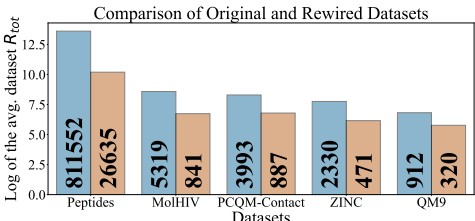

Figure 3: We compute the log of total effective resistance [Black et al., 2023] of five molecular datasets before and after rewiring the graphs using virtual nodes. Our rewiring technique consistently lowers the total effective resistance, indicating a better information flow on all of the datasets.

Table 4: Results on the ZINC [Jin et al., 2017] and OGBG-MOLHIV [Hu et al., 2020] datasets. Green is the best model, blue is the second, and red the third. The IPR-MPNN outperforms both SAT and GPS on ZINC, while obtaining the same performance as GPS on OGB-MOLHIV.

| MODEL | ZINC (12K) ↓ | OGB-MOLHIV ↑ |
|---|---|---|
| GINE [2019, 2023] | $0.101_{\pm 0.004}$ | $0.764_{\pm 0.010}$ |
| PR-MPNN [2023] | $0.084_{\pm 0.002}$ | $0.795_{\pm 0.009}$ |
| GPS [2022] | $0.070_{\pm 0.004}$ | $0.788_{\pm 0.010}$ |
| K-SG SAT [2022A] | $0.095_{\pm 0.002}$ | $0.613_{\pm 0.010}$ |
| K-ST SAT [2022A] | $0.115_{\pm 0.005}$ | $0.625_{\pm 0.039}$ |
| GRAPH MLP-MIXER [2023] | $0.073_{\pm 0.001}$ | $0.799_{\pm 0.010}$ |
| GRAPH VIT [2023] | $0.085_{\pm 0.005}$ | $0.779_{\pm 0.015}$ |
| IPR-MPNN (OURS) | $0.067_{\pm 0.001}$ | $0.788_{\pm 0.006}$ |

## 6 Conclusion

Here, we introduced implicit probabilistically rewired message-passing neural networks (IPR-MPNNs), a graph-rewiring approach leveraging recent progress in end-to-end differentiable sampling. IPR-MPNNs show drastically improved running times and memory usage efficiency over graph transformers and competing rewiring-based architectures due to IPR-MPNNs' ability to circumvent comparing every pair of nodes and significantly outperforming them on real-world datasets while effectively addressing over-squashing and overreaching. Hence, IPR-MPNNs represent a significant step towards designing scalable, adaptable MPNNs, making them more reliable and expressive.

**Acknowledgments**

CQ and CM are partially funded by a DFG (German Research Foundation) Emmy Noether grant (468502433) and RWTH Junior Principal Investigator Fellowship under Germany's Excellence Strategy. AM and MN acknowledge DFG funding under Germany's Excellence Strategy—EXC 2075 – 390740016, the support of the Stuttgart Center for Simulation Science (SimTech), and the International Max Planck Research School for Intelligent Systems (IMPRS-IS).

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

# A  Additional Related Work

In the following, we discuss related work.

**Graph Structure Learning**  Graph structure learning (GSL) is closely related to graph rewiring, where the primary motivation is refining and optimizing the graph structure while jointly learning graph representations [Zhou et al., 2023a]. Several methods have been proposed in this field. Jin et al. [2020] develop a technique to optimize graph structures from scratch using a specific loss function as a bias, while the general approach is using edge scoring functions for refinment  [Chen et al., 2020, Yu et al., 2021, Zhao et al., 2021], but discrete sampling methods have also been applied. More specifically, DGM [Kazi et al., 2022] is predicting the latent graph structure by leveraging Gumbel discrete sampling, while Franceschi et al. [2019] is learning a Bernoulli distribution via Hypergradient Descent. Saha et al. [2023] learns adaptive neighborhoods for trajectory prediction and point cloud classification by sampling through the smoothed-Heaviside function, while Younesian et al. [2023] samples nodes that are used for downstream tasks using GFlowNets. Some GSL techniques also employ unsupervised learning [Zou et al., 2023, Fatemi et al., 2021, Liu et al., 2022a,b]. We encourage the reader to refer to Fatemi et al. [2023], Zhou et al. [2023b] for detailed surveys regarding GSL.

Our proposed IPR-MPNN is different from other GSL framework in two main ways: (1) instead of sparsifying the graph using randomized $k$-NN approaches or independent Bernoulli random variables, we learn a probability mass function with exactly-$k$ constraints [Ahmed et al., 2023]. Moreover, we don't aim to discover the graph structure by considering a fully-connected latent graph from which we sample new edges [Qian et al., 2023], instead we introduce sparse connections from base nodes to virtual nodes, with complexity $N \cdot k$; (2) GSL methods do not investigate exact sampling of the exactly-$k$ distribution; however, one of our aims is to demonstrate that these techniques can significantly alleviate information propagation issues caused by inadequate graph connectivity, such as over-squashing and under-reaching.

Moreover, our IPR-MPNN also differs from previous graph rewiring method, specifically PR-MPNN [Qian et al., 2023]. First, we rewire the graph implicitly by connecting nodes to virtual nodes, respecting the original graph structure. Besides, we show a significant run-time advantage in that our worst-case complexity is sub-quadratic, while PR-MPNN optimally needs to consider $n^2$ node pairs for an $n$-order graph.

**Hierarchical MPNNs**  Our method draws connections to hierarchical MPNNs. The hierarchical model initially emerged in graph-level representation learning, as seen in approaches like AttPool [Huang et al., 2019], DiffPool  [Ying et al., 2018], and ASAP [Ranjan et al., 2020]. Further developments, such as Graph U-Net [Gao and Ji, 2019], H-GCN [Hu et al., 2019], and GXN [Li et al., 2020], introduced top-down and bottom-up methods within their architectures. However, they did not incorporate virtual node message passing. Other works create hierarchical MPNNs while incorporating inter-hierarchical message passing. For example, Fey et al. [2020] introduced HIMP-GNN on molecular learning, using a junction tree to create a higher hierarchy of the original graph and do inter-message passing between the hierarchies. Rampášek and Wolf [2021] proposed HGNet for long-range dependencies, generating hierarchies with edge pooling and training with relational GCN. Zhong et al. [2023] designed HC-GNN, more efficient than HGNet, for better node and higher level resolution community representations. These hierarchical MPNNs require well-chosen heuristics for hierarchy generation. Using an auxiliary supernode is particularly prominent in molecular tasks [Gilmer et al., 2017, Pham et al., 2017, Li et al., 2017], which involves adding a global node to encapsulate graph-level representations. Further advancements in this area, as suggested in [Battaglia et al., 2018], and developments like GWM Ishiguro et al. [2019], have enhanced supernode MPNNs with a gating mechanism. Theoretically, Cai et al. [2023] proves MPNN with a virtual node can simulate self-attention, which is further investigated in Rosenbluth et al. [2024]. In addition, Southern et al. [2024] studied the effect of MPNNs using a virtual node on over smoothing and oversquashing. Simultaneously, the recent study by Li et al. [2023] has introduced the concept of a collection of supernodes, termed "neural atoms," which incorporate supernode message passing with an attention mechanism. Moreover, the idea of a coarsened hierarchical graph has become widely employed in scalable MPNN training and graph representation learning, as evidenced by works like Huang et al. [2021], Liang et al. [2021], Namazi et al. [2022]. *Unlike existing hierarchical MPNNs, our IPR-MPNN uniquely leverages differential $k$-subset sampling techniques for dynamic, probabilistic graph rewiring and incorporates hierarchical message passing in an end-to-end trainable framework. This*

*approach enhances graph connectivity and expressiveness without relying on predefined heuristics or fixed structures.*

# B   Extended Notation

A *graph G* is a pair $(V(G), E(G))$ with *finite* sets of *vertices* or *nodes* $V(G)$ and *edges* $E(G) \subseteq \{\{u, v\} \subseteq V(G) \mid u \neq v\}$. If not otherwise stated, we set $n := |V(G)|$, and the graph is of *order n*. We also call the graph $G$ an *n-order* graph. For ease of notation, we denote the edge $\{u, v\}$ in $E(G)$ by $(u, v)$ or $(v, u)$. A *(vertex-)labeled graph G* is a triple $(V(G), E(G), \ell)$ with a (vertex-)label function $\ell : V(G) \to \mathbb{N}$. Then $\ell(v)$ is a *label* of $v$, for $v$ in $V(G)$. An *attributed graph G* is a triple $(V(G), E(G), \sigma)$ with a graph $(V(G), E(G))$ and (vertex-)attribute function $\sigma : V(G) \to \mathbb{R}^{1 \times d}$, for some $d > 0$. That is, contrary to labeled graphs, we allow for vertex annotations from an uncountable set. Then $\sigma(v)$ is an *attribute* or *feature* of $v$, for $v$ in $V(G)$. Equivalently, we define an *n-order* attributed graph $G := (V(G), E(G), \sigma)$ as a pair $\boldsymbol{G} = (G, \boldsymbol{L})$, where $G = (V(G), E(G))$ and $\boldsymbol{L}$ in $\mathbb{R}^{n \times d}$ is a *node feature matrix*. Here, we identify $V(G)$ with $[n]$. For a matrix $\boldsymbol{L}$ in $\mathbb{R}^{n \times d}$ and $v$ in $[n]$, we denote by $\boldsymbol{L}_{v.}$ in $\mathbb{R}^{1 \times d}$ the $v$th row of $\boldsymbol{L}$ such that $\boldsymbol{L}_{v.} := \sigma(v)$. Furthermore, we can encode an *n-order* graph $G$ via an *adjacency matrix* $\boldsymbol{A}(G) \in \{0, 1\}^{n \times n}$, where $A_{ij} = 1$ if, and only, if $(i, j) \in E(G)$. We also write $\mathbb{R}^d$ for $\mathbb{R}^{1 \times d}$.

The *neighborhood* of $v$ in $V(G)$ is denoted by $N(v) := \{u \in V(G) \mid (v, u) \in E(G)\}$ and the *degree* of a vertex $v$ is $|N(v)|$. Two graphs $G$ and $H$ are *isomorphic* and we write $G \simeq H$ if there exists a bijection $\varphi : V(G) \to V(H)$ preserving the adjacency relation, i.e., $(u, v)$ is in $E(G)$ if and only if $(\varphi(u), \varphi(v))$ is in $E(H)$. Then $\varphi$ is an *isomorphism* between $G$ and $H$. In the case of labeled graphs, we additionally require that $l(v) = l(\varphi(v))$ for $v$ in $V(G)$, and similarly for attributed graphs.

A *node coloring* is a function $c : V(G) \to \mathbb{R}^d$, $d > 0$, and we say that $c(v)$ is the *color* of $v \in V(G)$. A node coloring induces an *edge coloring* $e_c : E(G) \to \mathbb{N}$, where $(u, v) \mapsto \{c(u), c(v)\}$ for $(u, v) \in E(G)$. A node coloring (edge coloring) $c$ *refines* a node coloring (edge coloring) $d$, written $c \sqsubseteq d$ if $c(v) = c(w)$ implies $d(v) = d(w)$ for every $v, w \in V(G)$ $(v, w \in E(G))$. Two colorings are equivalent if $c \sqsubseteq d$ and $d \sqsubseteq c$, in which case we write $c \equiv d$. A *color class* $Q \subseteq V(G)$ of a node coloring $c$ is a maximal set of nodes with $c(v) = c(w)$ for every $v, w \in Q$. A node coloring is called *discrete* if all color classes have cardinality 1.

# C   The 1-dimensional Weisfeiler–Leman algorithm

The 1-WL or *color refinement* is a fundamental, well-studied heuristic for the graph isomorphism problem, originally proposed by Weisfeiler and Leman [1968].[1] The algorithm is an iterative method starting from labeling or coloring vertices in both graphs with degrees or other information, and updating the color of a node with its color as well as its neighbors' colors. During the iterations, two vertices with the same label get different labels if the number of identically labeled neighbors is unequal. Each iteration ends up with a vertex color partition, and the algorithm terminates when the partition is not refined by the algorithm, i.e., when a *stable coloring* or *stable partition* is obtained. We can finally conclude that the two graphs are not isomorphic if the color partitions are different, or the number of nodes of a specific color is different. Although in Cai et al. [1992] the limitation is shown that 1-WL algorithm cannot distinguish all non-isomorphic graphs, it is a powerful heuristic that can succeed on a broad class of graphs [Arvind et al., 2015, Babai and Kucera, 1979, Babai et al., 1980].

Formally, let $G = (V(G), E(G), \ell)$ be a labeled graph. In each iteration, $t > 0$, the 1-WL computes a vertex coloring $C_t^1 : V(G) \to \mathbb{N}$, depending on the coloring of the ego node and of the neighbors. That is, in iteration $t > 0$, we set

$$C_t^1(v) := \mathsf{RELABEL}\Big(\big(C_{t-1}^1(v), \{\!\{C_{t-1}^1(u) \mid u \in N(v)\}\!\}\big)\Big),$$

---

[1]Strictly speaking, the 1-WL and color refinement are two different algorithms. That is, the 1-WL considers neighbors and non-neighbors to update the coloring, resulting in a slightly higher expressive power when distinguishing vertices in a given graph; see Grohe [2021] for details. Following customs in the machine learning literature, we consider both algorithms to be equivalent.

for all vertices $v \in V(G)$, where RELABEL injectively maps the above pair to a unique natural number, which has not been used in previous iterations. In iteration 0, the coloring $C_0^1 := \ell$ is used. To test whether two graphs $G$ and $H$ are non-isomorphic, we run the above algorithm in "parallel" on both graphs. If the two graphs have a different number of vertices colored $c \in \mathbb{N}$ at some iteration, the 1-WL *distinguishes* the graphs as non-isomorphic. Moreover, if the number of colors between two iterations, $t$ and $(t+1)$, does not change, i.e., the cardinalities of the images of $C_t^1$ and $C_{i+t}^1$ are equal, or, equivalently,

$$C_t^1(v) = C_t^1(w) \iff C_{t+1}^1(v) = C_{t+1}^1(w),$$

for all vertices $v$ and $w$ in $V(G \mathbin{\dot\cup} H)$, then the algorithm terminates. For such $t$, we define the *stable coloring* $C_\infty^1(v) = C_t^1(v)$, for $v \in V(G \mathbin{\dot\cup} H)$. The stable coloring is reached after at most $\max\{|V(G)|, |V(H)|\}$ iterations [Grohe, 2017, Kiefer and McKay, 2020]. A function $f \colon V(G) \to \mathbb{R}^d$, for $d > 0$, is 1-WL-*equivalent* if $f \equiv C_\infty^1$.

## D  Additional Empirical Results and Experimental Details

Here, we provide additional empirical results and experimental details.

**Details**  In all of our real-world experiments, we use two virtual nodes with a hidden dimension twice as large as the base nodes. We randomly initialize the features of the virtual nodes. For the upstream and downstream models, we do a hyperparameter search; see Table A5. We use RWSE and LapPE positional encodings [Dwivedi et al., 2022a] for all of our experiments as additional node features, except for the synthetic TREES datasets [Alon and Yahav, 2021, Qian et al., 2023], EXP [Abboud et al., 2020] and CSL [Murphy et al., 2019], as well as node classification tasks on heterophilic datasets. For TREES-LEAFCOUNT, we use a single one-layer downstream GINE network, and for TREES-NEIGHBOURSMATCH, we use $n+1$ layers, where $n$ is the depth of the tree. For both TREES datasets, we have a hidden dimension of 32 for the original nodes. For most graph-level tasks, we apply a read-out function over the final pooled node embeddings, virtual node embeddings, or a combination of both. Distinctly, for PEPTIDES-FUNC, we apply read-out functions and use a supervised loss on all the intermediate embeddings, similarly to Errica et al. [2023]. We compute the logarithm of the symmetric sensitivity between the most distant two nodes $u, v$ in Fig. 2 similar to Errica et al. [2023], i.e. 1, and the total Effective Resistance of the datasets in Fig. 3 as in Black et al. [2023]. We optimize the network using Adam Kingma and Ba [2015] with a cosine annealing learning rate scheduler. We use the official dataset splits when available. Notably, for the TUDATASET, WEBKB datasets [Craven et al., 1998] and heterophilic datasets proposed in Platonov et al. [2023], we perform a 10-Fold Cross-Validation and report the average validation performance, similarly to the other methods that we compare with. For some experiments, we search for hyperparameters using grid-search, for more details please see Table A5. All experiments were performed on a mixture of A10, A100, A5000, and RTX 4090 NVIDIA GPUs. For each run, we used at most eight CPUs and 64 GB of RAM.

**Heterophilic datasets**  We exhibit experimental results on the WEBKB datasets in Table A6, and on the heterophilic datasets in Table A7. Our method exhibits significant improvement on WEBKB datasets as well as ROMAN-EMPIRE datasets compared with other MPNN baselines.

**Ablations**  We conduct ablation experiments on the number of virtual nodes and repetition of samples, see Table A8.

## E  Expressivity

We first discuss three scenarios where we (1) separate isomorphic graphs with the same non-discrete stable 1-WL colorings (Figure A5), (2) separate non-isomorphic graphs with the same non-discrete 1-WL colorings (Figure A6), and (3) preserve isomorphisms between isomorphic graphs with the same discrete 1-WL colorings (Figure A7). For each of the three examples, we assume two unique virtual nodes with $k = 1$, i.e., we sample a single edge between a base node and a virtual node.

For example (1), consider Figure A5 and assume that, for each color class, the upstream MPNN randomly selects a virtual node, i.e., we have a uniform prior. Since both base nodes have the same

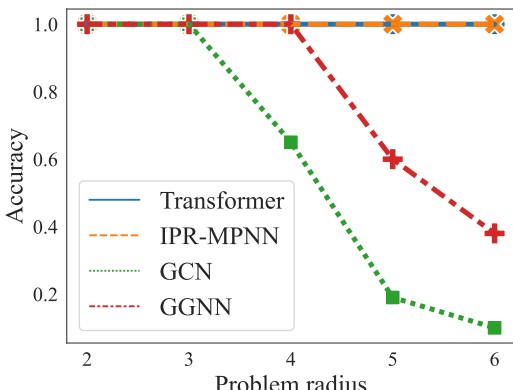

Figure A4: IPR-MPNN obtains perfect accuracy on TREES-NEIGHBORSMATCH [Alon and Yahav, 2021] for a depth up to 6.

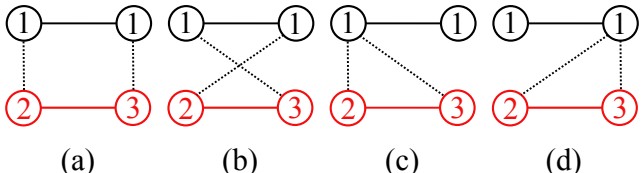

Figure A5: Possible new configurations.

color class, each base node connects to one of the virtual nodes uniformly at random. The scenario where two graphs remain isomorphic is when they connect to exactly the same virtual nodes or either connect as in case (a) or (b). Therefore, we have a high probability of separating these isomorphic graphs.

For example (2), we again produce uniform priors for each color class of the graphs in Figure A6. Once again, there is a high probability of separation. One possible configuration that can separate between the two graphs is shown in Figure A6 where in the first graph (a), the nodes colored with 1 get assigned to virtual node 4, while the nodes colored with 2 are assigned to virtual node 3. In the second graph (b), all nodes get assigned to the same virtual node 3.

For example (3) in Figure A7, we consider producing priors that assign, with high probability, the same virtual node to all of the nodes that are in color classes of cardinality 1. This approach ensures that the discretely-colored graphs remain isomorphic with high probability.

The intuition is that if we want to distinguish between graphs that have the same 1-$\mathsf{WL}$ stable color partitioning, the upstream model needs to produce "uninformative" prior weights for some color classes. However, preserving isomorphisms is most likely when the nodes in the same color class in the two graphs get assigned to the same virtual nodes. Since our upstream model is as powerful as 1-$\mathsf{WL}$, we can control the prior distribution for the color classes but not for individual nodes, therefore we can only guarantee high assignment probabilities for nodes in color classes of cardinality 1.

Next, we formally argue that we can preserve, with arbitrarily high probability, isomorphisms between graphs that have the same discrete stable 1-$\mathsf{WL}$ color partitions, as well as isomorphisms between subgraphs with the same discrete stable 1-$\mathsf{WL}$ color partitions.

**Lemma E.1.** *Let $G$ and $H$ be a pair of graphs with the same 1-$\mathsf{WL}$ graph coloring, i.e., they are 1-$\mathsf{WL}$ non-distinguishable. Let $k \in \mathbb{N}$, let $G'_k$ be* color-induced subgraph *where $V(G'_k) \coloneqq \{v \in V(G) \mid c^1_\infty(v) = k\} \subseteq V(G)$, and $V(H')$ similarly. Then the subgraphs induced by $V(G'_k)$ and $V(H'_k)$ are still not 1-$\mathsf{WL}$-distinguishable.*

*Proof.* To prove the lemma, we use the concept of an unrolling tree for a node; see, e.g., Morris et al. [2020b]. That is, for a node, we recursively unroll its neighbors, resulting in a tree. It is easy to see that two nodes get the same colors under 1-$\mathsf{WL}$ if and only if such trees are isomorphic; see Morris et al. [2020b] for details.

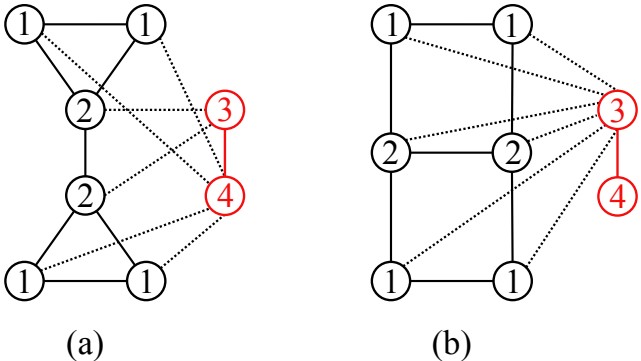

Figure A6: Separating graphs with the same stable 1-WL color.

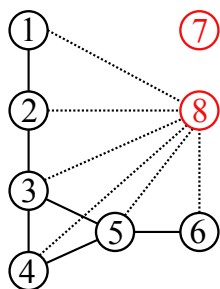

Figure A7: An example where we preserve the isomorphism when all the nodes from the initial graph are discretely colored. For each base node, we assign a high probability of connecting to the same virtual node (node 8).

Consider nodes $v, u \in V(G'_k) \cup V(H'_k)$ that share the same stable 1-WL coloring $k$. Based on the above, $v$ and $u$ must have isomorphic unrolling trees. Now remove subgraphs of the two trees not rooted at the vertex with color $k$. Since both $v$ and $u$ have isomorphic unrolling trees, the resulting trees are isomorphic. Hence, running 1-WL on top of $G'_k$ and $H'_k$ will still not distinguish them. $\square$

**Lemma E.2.** *Let $G$ and $H$ be a pair of graphs with the same stable coloring under 1-WL. If we add a finite number of virtual nodes on both graphs $C(G), C(H)$, and connect these virtual nodes based on 1-WL colors of the original graphs, i.e., two equally colored vertices get assigned the same virtual nodes. Then, the augmented graphs $\hat{G}$ and $\hat{H}$ have the same stable partition.*

*Proof.* The proof is by straightforward induction on the number of iterations using the fact that two nodes with the same color will be assigned to the same virtual node. That is, the neighborhood of two such nodes is extended by the same nodes.

$\square$

Besides, we leverage the following result by Qian et al. [2023], Morris et al. [2019].

**Lemma E.3** (Qian et al. [2023], Morris et al. [2019])**.** *Let $G$ be an $n$-order graph and let $c\colon V(G) \to \mathbb{R}^d$, $d > 0$, be a 1-WL-equivalent node coloring. Then, for all $\varepsilon > 0$, there exists a (permutation-equivariant) MPNN $f\colon V(G) \to \mathbb{R}^d$, such that*

$$\max_{v \in V(G)} \|f(v) - c(v)\| < \varepsilon.$$

**Theorem E.4.** *Let $k > 0$, $\varepsilon \in (0, 1)$, and $G, H$ be two graphs with identical 1-WL stable colorings. Let $M$ be the set of ordered virtual nodes, $V_G$ and $V_H$ be the subset of nodes in $G$ and $H$ that have a color class of cardinality 1, with $|V_G| = |V_H| = d$, and $W_G$, $W_H$ the subset of nodes that have a color class of cardinality greater than 1, with $|W_G| = |W_H| = n$. Then, for all choices of 1-WL-equivalent functions $f$,*

*(1) there exists a conditional probability mass function $p_{(\boldsymbol{\theta},k)}$ that does* not *separate $G[V_G]$ and $H[V_H]$ with probability at least $1 - \varepsilon$.*

*(2) There exists a conditional probability mass function $p_{(\boldsymbol{\theta},k)}$ that separates $G[W_G]$ and $H[W_H]$ with probability strictly greater than $0$.*

*Proof.* Let $M = \{v_1, ..., v_m\}$ be the set of $m$ ordered virtual nodes and $d = |V_G| = |V_H|$. To prove this theorem, we can leverage Lemma E.3 to assign distinct and arbitrary priors for every color class.

For $(1)$, we know that since $G$ and $H$ have identical 1-WL stable colorings and $V_G$, $V_H$ have a color class of cardinality 1, then $G[V_G]$ and $H[V_H]$ must be discrete and isomorphic. Using Lemma E.3, we can obtain an upstream MPNN that assigns a sufficiently high prior $\theta_i$ such that, when we sample from the exactly-$k$ distribution, we can assign the corresponding $k$ virtual nodes to a base node with probability at least $\sqrt[2d]{1 - \varepsilon}$.

To demonstrate the existence of such a set of priors $\boldsymbol{\theta}$, let $\delta \in (0, 1)$ and $S \subset M$ be a subset of $k$ virtual nodes. Let $w_1 > w_2$ be two prior weights such that $\theta_i = w_1$ if $v_i \in S$, and $\theta_i = w_2$ if $v_i \in M \setminus S$. We have that

$$p_{\theta,k}(S) \geq \delta \left( \sum_{i=0}^{k} \binom{k}{i} \binom{m-k}{k-i} w_1^i w_2^{k-i} \right) = \delta Z,$$

with the upper bound [Qian et al., 2023]

$$Z \leq w_1^k + \left( \binom{m}{k} - 1 \right) w_2 w_1^{k-1},$$

Thus, $\boldsymbol{\theta}$ exists and can be obtained using this inequality.

Next, we set $\delta = \sqrt[2d]{1 - \varepsilon}$. Consequently, the probability that the sampled virtual nodes are identical for both graphs is at least $\sqrt[2d]{1 - \varepsilon}^{2d} = 1 - \varepsilon$. Finally, using Lemma E.2, we know that the two graphs also retain their color partitions and remain isomorphic with probability at least $1 - \varepsilon$.

For $(2)$, it is easy to see that, for any prior weights $\boldsymbol{\theta}$ that we assign to the color classes of cardinality greater than 1, there is at least one configuration separating the two graphs. For instance, since $k < m$, we can separate $G[W_G], H[W_H]$ by assigning the first $k$ virtual nodes $\{1, ..., k\}$ to every node in $G[W_G]$, but have at least one node in $H[W_H]$ be assigned to the next $k$ nodes $\{2, ..., k+1\}$. More concretely, let $\varepsilon \in (0, 1)$, $v \in G[W_G] \cup H[W_H]$ and $\boldsymbol{\theta}_u$ be an uniform prior, i.e. $\boldsymbol{\theta}_1 = \boldsymbol{\theta}_2 = ... = \boldsymbol{\theta}_m$. Again, we use Lemma E.3 and obtain a distribution $p_{\boldsymbol{\theta},k}$, arbitrarily close to the uniform distribution. Then, the probability of making the two graphs distinguishable by obtaining the mentioned example is greater than $\frac{1-\varepsilon}{\binom{m}{k}^{2n}}$, which is strictly greater than 0. For a visual example, see Figure A6. $\qquad\square$

The next Corollary follows directly from Theorem E.4, and recovers Theorem 4.1 from Qian et al. [2023].

**Corollary E.4.1.** *For sufficiently large $n$, for every $\varepsilon \in (0, 1)$, a set $M$ of ordered virtual nodes, and $k > 0$, we have that almost all pairs, in the sense of Babai et al. [1980], of isomorphic n-order graphs $G$ and $H$ and all permutation-invariant, 1-WL-equivalent functions $f : \mathfrak{A}_n \to \mathbb{R}^d$, $d > 0$, there exists a probability mass function $p_{(\boldsymbol{\theta},k)}$ that separates the graph $G$ and $H$ with probability at most $\varepsilon$ with respect to $f$.*

*Proof.* We know from Babai et al. [1980] that an 1-WL-equivalent algorithm will produce a discrete color partition for almost all pairs of isomorphic graphs $G$, $H$ of sufficient size. We use Theorem E.4 and set $W_G = W_H = \emptyset$ and conclude that we maintain isomorphisms between almost all isomorphic graphs. $\qquad\square$

## F   Limitations

A limitation of our approach is the assumption that the number of virtual nodes $m$ is significantly smaller than the total number of nodes $n$. As the number of virtual nodes increases, the runtime is also expected to rise (see Table A12 for a detailed example). In the worst-case scenario, where $m = n$, our method exhibits quadratic complexity. However, in all real-world datasets we have tested, the required number of virtual nodes for achieving optimal performance is low. For more information, refer to Table A5. Another question is whether IPR-MPNNs can perform well on node-level tasks. We have designed our rewiring method specifically to solve long-range graph-level tasks (such as the tasks on the molecular datasets from the LONG-RANGE GRAPH BENCHMARK [Dwivedi et al., 2022b]). Nevertheless, IPR-MPNNs and adaptations might also work on node-level tasks, but we leave this question open for further work.

Table A5: Overview of used hyperparameters.

| DATASET | HIDDEN_UPSTREAM | LAYERS_UPSTREAM | HIDDEN_DOWNSTREAM | HIDDEN_VIRTUAL | LAYERS_DOWNSTREAM | K | VIRTUAL NODES | SAMPLES_TRAIN/TEST |
|---|---|---|---|---|---|---|---|---|
| ZINC | 128 | 2 | 128 | 256 | 10 | 1 | 2 | 2 |
| OGBG-MOLHIV | {64,128} | {0,2,5} | {64,128,256} | {64,128,256} | {3,5,8} | 1 | 2 | 2 |
| QM9 | 128 | 2 | 128 | 256 | 10 | 1 | 2 | 2 |
| PEPTIDES-FUNC | 128 | 2 | 128 | 256 | 10 | 1 | 2 | 2 |
| PEPTIDES-STRUCT | 128 | 2 | 128 | 256 | 10 | 1 | 2 | 2 |
| MUTAG | {64,128} | {0,2,5} | {64,128} | {64,128} | {3,5,8} | 1 | 2 | 2 |
| PTC_MR | {64,128} | {0,2,5} | {64,128} | {64,128} | {3,5,8} | 1 | 2 | 2 |
| NCI1 | {64,128} | {0,2,5} | {64,128} | {64,128} | {3,5,8} | 1 | 2 | 2 |
| NCI109 | {64,128} | {0,2,5} | {64,128} | {64,128} | {3,5,8} | 1 | 2 | 2 |
| PROTEINS | {64,128} | {0,2,5} | {64,128} | {64,128} | {3,5,8} | 1 | 2 | 2 |
| TREES-LEAFCOUNT | 32 | 2 | 32 | 64 | 1 | 1 | 2 | 2 |
| TREES-LEAFMATCH-N | 32 | 2 | 32 | 64 | {N+1} | 7 | 2 | 2 |
| CSL | 64 | 1 | 64 | 64 | 6 | 7 | 8 | 15 |
| EXP | 64 | 1 | 64 | 128 | 6 | 3 | 4 | 2 |
| CORNELL | 128 | 2 | 256 | 284 | 1 | 1 | 2 | 2 |
| TEXAS | 128 | 3 | 256 | 284 | 3 | 1 | 2 | 2 |
| WISCONSIN | 64 | 3 | 128 | 284 | 1 | 1 | 2 | 3 |
| ROMAN-EMPIRE | 64 | 2 | 256 | 256 | 3 | 1 | 3 | 1 |
| TOLOKERS | 128 | 3 | 256 | 384 | 3 | 1 | 1 | 1 |
| MINESWEEPER | 64 | 2 | 256 | 256 | 3 | 1 | 3 | 1 |
| AMAZON-RATINGS | 128 | 2 | 256 | 128 | 4 | 1 | 3 | 3 |

Table A6: Performance comparison of different models on the Cornell, Texas, and Wisconsin heterophilic datasets.

| MODEL | CORNELL ↑ | TEXAS ↑ | WISCONSIN ↑ |
|---|---|---|---|
| GINE 2019 | $0.448_{\pm0.073}$ | $0.650_{\pm0.068}$ | $0.517_{\pm0.054}$ |
| SDRF 2021 | $0.546_{\pm0.004}$ | $0.644_{\pm0.004}$ | $0.555_{\pm0.003}$ |
| DIGL 2019 | $0.582_{\pm0.005}$ | $0.620_{\pm0.003}$ | $0.495_{\pm0.003}$ |
| GEOM-GCN 2020 | $0.608_{\pm N/A}$ | $0.676_{\pm N/A}$ | $0.641_{\pm N/A}$ |
| DIFFWIRE 2022 | $0.690_{\pm0.044}$ | N/A | $0.791_{\pm0.021}$ |
| GRAPHORMER 2021 | $0.683_{\pm0.017}$ | $0.767_{\pm0.017}$ | $0.770_{\pm0.019}$ |
| GPS 2022 | $0.718_{\pm0.024}$ | $0.773_{\pm0.013}$ | $0.798_{\pm0.090}$ |
| IPR-MPNN (OURS) | $\mathbf{0.764}_{\pm0.056}$ | $\mathbf{0.808}_{\pm0.052}$ | $\mathbf{0.804}_{\pm0.052}$ |

Table A7: Performance comparison between the base GINE and IPR-MPNN on recently-proposed heterophilic datasets.

| MODEL | ROMAN-EMPIRE | TOLOKERS | MINESWEEPER | AMAZON-RATINGS |
|---|---|---|---|---|
| GINE (BASE) | $0.476_{\pm0.006}$ | $0.807_{\pm0.006}$ | $0.799_{\pm0.002}$ | $\mathbf{0.488}_{\pm0.006}$ |
| IPR-MPNN (OURS) | $\mathbf{0.839}_{\pm0.006}$ | $\mathbf{0.820}_{\pm0.008}$ | $\mathbf{0.887}_{\pm0.006}$ | $0.480_{\pm0.007}$ |

Table A8: Performance between a virtual node connected to the entire original graph (1VN-FC) and IPR-MPNNs with two virtual nodes with one sample (2VN1S) and two samples, respectively (2VN2S).

| MODEL | ZINC | OGB-MOLHIV | PEPTIDES-FUNC | PEPTIDES-STRUCT |
|---|---|---|---|---|
| 1VN - FC | $0.074_{\pm0.002}$ | $0.753_{\pm0.011}$ | $0.7039_{\pm0.0046}$ | $0.2435_{\pm0.0007}$ |
| 2VN1S | $0.072_{\pm0.004}$ | $0.762_{\pm0.014}$ | $0.7146_{\pm0.0055}$ | $0.2472_{\pm0.0014}$ |
| 2VN2S | $\mathbf{0.067}_{\pm0.001}$ | $\mathbf{0.788}_{\pm0.006}$ | $\mathbf{0.7210}_{\pm0.0039}$ | $\mathbf{0.2422}_{\pm0.0007}$ |
| 2VN4S | – | – | $0.7145_{\pm0.0020}$ | – |

Table A9: IPR-MPNN compared to other approaches as reported in Giusti et al. [2023b], Karhadkar et al. [2022], Papp et al. [2021], Arnaiz-Rodríguez et al. [2022], Qian et al. [2023]. **Green** indicates the best model, **blue** the second-best, and **red** the third. Our rewiring technique obtains the best performance on every dataset, except for MUTAG, where PR-MPNN obtains a slightly better average mean score and standard deviation.

| MODEL | MUTAG | PTC_MR | PROTEINS | NCI1 | NCI109 |
|---|---|---|---|---|---|
| GK ($k = 3$) [2009] | $81.4_{\pm1.7}$ | $55.7_{\pm0.5}$ | $71.4_{\pm0.3}$ | $62.5_{\pm0.3}$ | $62.4_{\pm0.3}$ |
| PK [2016] | $76.0_{\pm2.7}$ | $59.5_{\pm2.4}$ | $73.7_{\pm0.7}$ | $82.5_{\pm0.5}$ | N/A |
| WL KERNEL [2011] | $90.4_{\pm5.7}$ | $59.9_{\pm4.3}$ | $75.0_{\pm3.1}$ | $86.0_{\pm1.8}$ | N/A |
| DGCNN [2018] | $85.8_{\pm1.8}$ | $58.6_{\pm2.5}$ | $75.5_{\pm0.9}$ | $74.4_{\pm0.5}$ | N/A |
| IGN [2019B] | $83.9_{\pm13.0}$ | $58.5_{\pm6.9}$ | $76.6_{\pm5.5}$ | $74.3_{\pm2.7}$ | $72.8_{\pm1.5}$ |
| GIN [2019] | $89.4_{\pm5.6}$ | $64.6_{\pm7.0}$ | $76.2_{\pm2.8}$ | $82.7_{\pm1.7}$ | N/A |
| PPGNS [2019A] | $90.6_{\pm8.7}$ | $66.2_{\pm6.6}$ | $77.2_{\pm4.7}$ | $83.2_{\pm1.1}$ | $82.2_{\pm1.4}$ |
| NATURAL GN [2020] | $89.4_{\pm1.6}$ | $66.8_{\pm1.7}$ | $71.7_{\pm1.0}$ | $82.4_{\pm1.3}$ | $83.0_{\pm1.9}$ |
| GSN [2022] | $92.2_{\pm7.5}$ | $68.2_{\pm7.2}$ | $76.6_{\pm5.0}$ | $83.5_{\pm2.0}$ | $83.5_{\pm2.3}$ |
| CIN [2021] | $92.7_{\pm6.1}$ | $68.2_{\pm5.6}$ | $77.0_{\pm4.3}$ | $83.6_{\pm1.4}$ | $84.0_{\pm1.6}$ |
| CAN [2023A] | $94.1_{\pm4.8}$ | $72.8_{\pm8.3}$ | $78.2_{\pm2.0}$ | $84.5_{\pm1.6}$ | $83.6_{\pm1.2}$ |
| CIN++ [2023B] | $94.4_{\pm3.7}$ | $73.2_{\pm6.4}$ | $80.5_{\pm3.9}$ | $85.3_{\pm1.2}$ | $84.5_{\pm2.4}$ |
| GT [2020] | $83.9_{\pm6.5}$ | $58.4_{\pm8.2}$ | $70.1_{\pm3.2}$ | $80.0_{\pm1.9}$ | N/A |
| GRAPHIT [2021] | $90.5_{\pm7.0}$ | $62.0_{\pm9.4}$ | $76.2_{\pm4.4}$ | $81.4_{\pm2.2}$ | N/A |
| FOSR [2022] | $86.2_{\pm1.5}$ | $58.5_{\pm1.7}$ | $75.1_{\pm0.8}$ | $72.9_{\pm0.6}$ | $71.1_{\pm0.6}$ |
| DROPGNN [2021] | $90.4_{\pm7.0}$ | $66.3_{\pm8.6}$ | $76.3_{\pm6.1}$ | $81.6_{\pm1.8}$ | $80.8_{\pm2.6}$ |
| GAP(R) [2022] | $86.9_{\pm4.0}$ | N/A | $75.0_{\pm3.0}$ | N/A | N/A |
| GAP(N) [2022] | $86.9_{\pm4.0}$ | N/A | $75.3_{\pm2.1}$ | N/A | N/A |
| PR-MPNN [2023] | $98.4_{\pm2.4}$ | $74.3_{\pm3.9}$ | $80.7_{\pm3.9}$ | $85.6_{\pm0.8}$ | $84.6_{\pm1.2}$ |
| IPR-MPNN (OURS) | $98.0_{\pm3.4}$ | $75.8_{\pm5.3}$ | $85.4_{\pm4.4}$ | $86.2_{\pm1.2}$ | $86.5_{\pm1.4}$ |

Table A10: Comparison between the base GIN model, its variants, and IPR-MPNN on the EXP dataset.

| MODEL | ACCURACY ↑ |
|---|---|
| GIN | $0.511 \pm 0.021$ |
| GIN + ID-GNN | $1.000 \pm 0.000$ |
| PR-MPNN | $1.000 \pm 0.000$ |
| IPR-MPNN (OURS) | $1.000 \pm 0.000$ |

Table A11: Comparison between the base GIN model w/wo positional encoding and IPR-MPNN on CSL dataset. For IPR-MPNN*, we pre-calculate the graph partitioning for each data instance, and label each node with its partition ID.

| MODEL | ACCURACY ↑ |
|---|---|
| GIN | $0.100 \pm 0.000$ |
| GIN + POSENC | $1.000 \pm 0.000$ |
| PR-MPNN | $0.998 \pm 0.008$ |
| IPR-MPNN (OURS) | $0.987 \pm 0.013$ |
| IPR-MPNN$^*$ (OURS) | $1.000 \pm 0.000$ |

Table A12: More memory consumption details together with train and validation times per epoch in seconds. We compare to the base GINE model, various variants of the SAT Graph Transformer, GraphGPS, and the PR-MPNN rewiring technique. IPR-MPNNs maintain low memory usage while also being significantly faster when compared to the Graph Transformers and PR-MPNN. The experiments were performed on the OGBG-MOLHIV dataset, with the same batch size and the same machine that contains an Nvidia RTX A5000 GPU and an Intel i9-11900K CPU.

| MODEL | #PARAMS | V. NODES | SAMPLES | TRAIN S/EP | VAL S/EP | MEM. USAGE |
|---|---|---|---|---|---|---|
| GINE | $502k$ | - | - | $3.19 \pm 0.03$ | $0.20 \pm 0.01$ | 0.5GIB |
| K-ST SAT$_{\text{GINE}}$ | $506k$ | - | - | $86.54 \pm 0.13$ | $4.78 \pm 0.01$ | 11.0GIB |
| K-SG SAT$_{\text{GINE}}$ | $481k$ | - | - | $97.94 \pm 0.31$ | $5.57 \pm 0.01$ | 8.5GIB |
| K-ST SAT$_{\text{PNA}}$ | $534k$ | - | - | $90.34 \pm 0.29$ | $4.85 \pm 0.01$ | 10.1GIB |
| K-SG SAT$_{\text{PNA}}$ | $509k$ | - | - | $118.75 \pm 0.50$ | $5.84 \pm 0.04$ | 9.1GIB |
| GRAPHGPS | $558k$ | - | - | $17.02 \pm 0.70$ | $0.65 \pm 0.06$ | 6.6GIB |
| PR-MPNN$_{\text{GMB}}$ | $582k$ | - | 20 | $15.20 \pm 0.08$ | $1.01 \pm 0.01$ | 0.8GIB |
| PR-MPNN$_{\text{IMLE}}$ | $582k$ | - | 20 | $15.01 \pm 0.22$ | $1.08 \pm 0.06$ | 0.9GIB |
| PR-MPNN$_{\text{SIM}}$ | $582k$ | - | 20 | $15.98 \pm 0.13$ | $1.07 \pm 0.01$ | 2.1GIB |
| IPR-MPNN$_{\text{SIM}}$ | $548k$ | 2 | 1 | $7.31 \pm 0.08$ | $0.34 \pm 0.01$ | 0.8GIB |
| IPR-MPNN$_{\text{SIM}}$ | $548k$ | 4 | 2 | $7.37 \pm 0.08$ | $0.35 \pm 0.01$ | 0.8GIB |
| IPR-MPNN$_{\text{SIM}}$ | $548k$ | 10 | 5 | $7.68 \pm 0.10$ | $0.35 \pm 0.01$ | 0.9GIB |
| IPR-MPNN$_{\text{SIM}}$ | $549k$ | 20 | 10 | $8.64 \pm 0.06$ | $0.35 \pm 0.01$ | 1.1GIB |
| IPR-MPNN$_{\text{SIM}}$ | $549k$ | 30 | 20 | $9.41 \pm 0.38$ | $0.43 \pm 0.01$ | 1.2GIB |

