# OpenReview forum: "Probabilistic Graph Rewiring via Virtual Nodes"
_NeurIPS.cc/2024/Conference — NeurIPS 2024 poster_

### Official Review · Reviewer_5s4E · 2024-06-15

**Soundness:** 3
**Presentation:** 2
**Contribution:** 3
**Rating:** 5
**Confidence:** 5

**Summary:**

This work introduces an approach called implicitly rewired message-passing neural networks (IPR-MPNNs) to address limitations in message-passing graph neural networks (MPNNs): expressiveness and mainly over-squashing. Their method works by adding virtual nodes and learning to connect them to existing nodes in a differentiable, end-to-end manner. Authors claim that previous techniques are not scalable, and state that their approach is more scalable. They empirically show the performance of their proposed model.

**Strengths:**

* Their method slighlty improves the computational complexity when using m<<n virtual nodes (otherwise is quadratic as well).
* Empirically, their proposed method improves the accuracy in some of the datasets wrt some of the baselines.
* Theoretically, IPR-MPNNs exceed the expressiveness of MPNNs, overcoming 1WL test.
* The method allows for end-to-end differentiable learning, integrating the rewiring process seamlessly into the training of MPNNs. This is done by few previous approaches (see next section)


**[POST-REBUTTAL COMMENT]**

After carefully reading the answer from the authors, I acknowledge the improvement of the manuscript after the rebuttal (related work, additional discussion, clarification of misunderstandings, additional experiments...), I increase my score and I support the acceptance of the manuscript.

**Weaknesses:**

### Related work

* The state of the art is in general not properly structured and not enough detailed. Although they cite some of the important work, thay do not delve into the detail of the different papers about limitations on oversquashing (Di Giovanni 2023, Black 2023, they cite both of them, but authors dot seem to cite them when talking about limitations or even they dont mention the rewiring methods they propose SDRF and GTR), which is one of the main points in this work. For instance, when talking about graph rewiring they cite several works, no detailing what are the contributions of each and also mixing works of different solutions in the same sentence: MixHop is based on a adaptative message passing and DIGL is rewiring basd on difussion, but authors just say that both works are "considered as graph rewiring as they can reach further-away neighbors in a single layer". I mean, authors mix adaptative MP with spatial rewiring with spectral rewiring, expanders with no explanation about it. I do not think that the level of structure and detail is enough for a paper in this conference. See Di giovanni 2023, Banerjee 2022a(FOSR), 2022b(RLEF), Choi 2024(Pandas), for an example of better structured and detailed explanation of the different rewiring methods.

* In addition, they miss very relevant works in the literature about graph rewiring and the analysis of graph limitations that are related with this paper. Specifically, authors evaluate their contribution using commute time or effective resistance. They miss 2 very important rewiring works in this line DiffWire (Arnaiz-Rodriguez, 2022) and FOSR (Banerjee, 2022), where they introduce the oversquashing connectino with the Cheger constant and methods to reduce the effective resistance between nodes to overcome oversquashing. To be even more clear regarding Banerjee 2022: in the references appears RLEF and FOSR, but in the text there is only one reference to RLEF. They also do not mention GTR from the already cited paper Black 2023, which is also based on effective resistance, metric that authors then use to evaluate IPR-MPNN. I suggest the authors to look more carefully also Di Giovanni 2023 on all the impact of not only these but also more works in the graph rewiring literature, and to completely cite all the important first works in this area.

* On top of DiffWire and FOSR, authors also miss more important papers in grph rewiring such as Drew (Gutteridge 2023), LASER (Barbero 2024) or Affinity-Aware GNNs (Velingker).

* In addition, they miss relevant papers in the literature about virtual nodes, which is one of the main points in this paper. For instance, the two first important works about virtual nodes (Battaglia 2017 and Brüel-Gabrielsson 2022) are just cited in the appendix and/or not even mentioning that they proposed the virtual nodes. Additionally, authors are not aware of works focused on the limitations of virtual nodes and how they are related with oversquashing: Southern 2024 and Geisler 2024. These papers completelly address how *theoretically* virtual nodes affect oversquashing and how they can be used to mitigate it (see methodology weaknesses). I suggest the authors to look more carefully at the literature about virtual nodes and oversquashing, and to completely and properly cite all the important first works in this area correctly stating the main contributions of the prior work.


### Methodology

* The contribution of this work is limited since it comes from the combination of virtual nodes and probabilistic message passing. Both ideas were already previously proposed. It would be helpful if authors are more transparent on what are the exact novelty of this method. Virtual nodes ar not novelty as proposed before (Battaglia 2017) and probabilitistic message passing as well. The changes to make this approach novel are not sufficient in my opinion.

* Regarding the contributions, only the expressiveness contribution is shown theoretically, and it is the less related with the problem they claim to solve (OVSQ). The rest of the contributions are just shown empirically, which does not advance on the knowledge of the limitations of the GNN field.
* Authors do not justify thoretically how their approach improves oversquashing, which should be the main approach of the paper. They just empirically show that the effective resitance and sensitivity are smalles, which is a trivial result of the Rayleigh’s monotonicity principle when densifying a graph. The paper would increase a lot the quality if authors really tryy to understand why their method improve oversauashing and if it really does. Souther 2024 and Geisler 2024 might be helpful for this. To reach the level of Neurips authors should have a more theoretical approach to the problem they are trying to solve. However, it is monstly empirical as they also aknowledge in the section 5 (they answer all the questions just empirically). This is very dangerous, since the empirical results might not be generalizable to other datasets or tasks. Also, in the GNN community is widely known the lack of understanding of how GNNs work if we only look at the results of datasets.


* One key contribution is that they do rewiring it in less than quadratic complexity (which is only ok for transphormers but is not ok for rewiring methods). However this is only empirically, since authors also mention l865 "In the worst-case scenario, where m = n, our method exhibits quadratic complexity. However, in all real-world datasets we have tested, the required number of virtual nodes for achieving optimal performance is low".

* Also, mention do not delve into the limitation of the virtual and oversquashing (See Southern 2024 and Geisler 2024). For instance, altough the effective resistance is reduced using virtual nodes, the oversquashing is not fully solved since all the messages are squeezed in the virtual nodes, even worsening the problem of exponential aggregation of information in some bottleneck "nodes". As shown in Topping 2022 and Arnaiz-Rodriguez 2022, the blottleneckness of the nodes (measured either by curvature, spectral metrics or centralities), causes oversquashing.

* As their main experiments are in graph classification tasks,  and the main contribution is based on pairwise scores, they also should compare aagaints CT-Layer in Diffwire (Arnáiz-Rodrigues 2023), which is a rewiring based also in similarity scores that was shown to improve graph classification tasks. Also, authors claim that previous "strategies to mitigate over-squashing rely on heuristic rewiring methods or purely randomized approaches that may not adapt well to a given prediction task". Also, diffwire is in-processing rewiring and takes into account the task. Finally, also Geisler 2024 is task-oriented.


* They compare the running time against transformer but not against rewiring, which does not make sense. They propose an architecture to do graph rewiring and alliviate oversquashing, so they should compare with graph rewiring methods.


### Refs

P. W. Battaglia,  et al. Relational inductive biases, deep learning, and graph networks. 2017.

A. Arnaiz-Rodriguez et al. DiffWire: Inductive Graph Rewiring via the Lovász Bound. LoG, 2022.

P. K. Banerjee et al. Oversquashing in gnns through the lens of information contraction and graph expansion. AACCC, 2022.

R. Brüel-Gabrielsson, M. Yurochkin, and J. Solomon. Rewiring with positional encodings for graph neural networks. 2022

F. Di Giovanni et al. On over-squashing in message passing neural networks: The impact of width, depth, and topology. ICML 2023.

M. Black et al. Understanding Oversquashing in GNNs through the Lens of Effective Resistance. ICML 2023.

D. Gutteridge et al. Drew: Dynamically rewired message passing with delay. ICML 2023.

A. Velingker. Affinity-Aware Graph Networks. NeurIPS 2023.

F. Barbero. Locality-Aware Graph Rewiring in GNNs. ICML 2024.

J. Choi. PANDA: Expanded Width-Aware Message Passing Beyond Rewiring. ICML 2024

J. Southern. Understanding Virtual Nodes: Oversmoothing, Oversquashing, and Node Heterogeneity. 2024.

S. Geisler. Spatio-Spectral Graph Neural Networks. 2024.

**Questions:**

Questions are listed along with each specific identified weaknesses in the previous section.

**Limitations:**

I summarize the limitations of the work, however, I refer to weaknesses section for a more detailed feedback.

* Relate work is not complete and not clearly explained and structured.
* The authors do not closely connect how virtual nodes solve oversquahing.
* Authors do not discuss limitations of the virtual nodes wrt over-squashing.
* All the important contributions are empirical, and the theoretical ones are not very novel.
* The core of the idea is not very novel, just a combination of previous work on virtual nodes an probabilistic message passing, but not a big contribution by itself. Also, they mention as contributions that their methods alliviates the limitations of GNNs, but there is no explanation why, just empirical results.

---

> ### Author Rebuttal · Authors · 2024-08-06
>
> We thank the reviewer for their feedback and detailed review.
>
> Due to the space restrictions for the rebuttal, we address related work and novelty comments here, with the remaining concerns, additional results, and the bibliography in other official comments at the same level.
>
> - **Q: Related work.**
>
> - **R:** We thank the reviewer for the feedback on our related work sections. We agree that giving a comprehensive overview of current literature is essential. Based on the reviewer's comments, we will improve the related work sections for the final paper.
>
> In the following, we discuss the related work that the reviewer has proposed in detail:
>
> **Missing concurrent related work:**
>
> * **[SN24, GR24]: We would like to point out to the reviewer and the program committee that these are concurrent works that became publicly available on arXiv on 22 May and 29 May, respectively. The NeurIPS Main Conference Paper Submission deadline was 22 May, AoE, so it would have been impossible to include the papers in the related work section at submission time.**
>
> ---
>
> **Related work proposals that have been discussed:**
>
> * **[GE23]:** This is a key work we compare with, citing and discussing it on lines 35, 101-102, and 253. We compare with Drew on the QM9 dataset (Table 2), where we outperform it on 12 out of 13 properties, and on Peptides-func and Peptides-struct (Table 3), where we outperform it on both datasets.
>
> * **[BGN22]:** When discussing related rewiring approaches, the first work we mention is [BGN22] (lines 97-98).
>
> * **[BA17]:**  We include related work regarding Hierarchical MPNNs in our Supplementary Materials. We provide an extensive discussion on both hierarchical methods and virtual node-based methods, including Battaglia 2017 (lines 677-678).
>
> * **[BE23]:** We cite this method on line 106.
>
> * **[KR22]:** We cite this work on line 35 and briefly discuss it on line 105. We also compare with their FoSR method on the TUDataset collection in Table A6, where we achieve significantly better results on all datasets.
>
> ---
>
> **Missing related work:**
>
> * **[ARZ22, VR23, BO24]:** We thank the reviewer for pointing out that this related work is missing.
>
> We will update our manuscript to include the concurrent works of [SN24, GR24] and comparisons with them on the peptides datasets. We will add the missing related work of [ARZ22, VR23, BO24] and expand the discussions on [BE23, KR22].
>
> ---
>
> * **Q: Methodology.**
>
> * **R:** We thank the reviewer for their suggestions; in the following, we try to respond to their concerns:
>
> 1. **Novelty:** While probabilistic rewiring [QN24] and the concept of virtual nodes [BA17] have been explored in the past, combining the two concepts is not trivial. As highlighted in our paper, the key difference between the present work and [QN24] is that, unlike [QN24], where the adjacency matrix is modified directly, we keep the original adjacency matrix fixed and "rewire" by learning a $k$-subset distribution from the original graph nodes to the virtual nodes. Moreover, the virtual nodes can have different dimensions, connecting our message-passing scheme to heterogeneous message-passing and hierarchical MPNNs. Unlike existing hierarchical MPNNs that employ virtual/super nodes, e.g., [BA17, PM17, LI17, CI23], our IPR-MPNNs uniquely leverage differential $k$-subset sampling and incorporate hierarchical message-passing in an end-to-end framework. We discuss our connection to hierarchical message-passing and virtual nodes-based methods in the Supplementary materials, lines 663-689.
>
> 2. **Theoretical guarantees:** Our primary goal is to improve the long-range capabilities of MPNNs while keeping complexity, memory usage, and runtimes low. Long-range dependency modeling is affected by over-squashing and under-reaching, but they are not our core theoretical concern. In our work, the theoretical guarantees of expressiveness, along with the over-squashing and under-reaching analyses, indicate where the performance gains might come from.
>
>
> 3. **Empirical results might not be generalizable:** Our experimental section offers a comprehensive comparison on both real-world molecular datasets (ZINC, OGBG-Molhiv, QM9, Peptides-struct, Peptides-func, PCQM-Contact, TUDataset) and synthetic datasets (Trees-LeafCount, Trees-LeafMatch, CSL, EXP), where we generally achieve state-of-the-art results. Our empirical evidence shows that our method is effective in addressing over-squashing through model sensitivity (Fig. 2), effective resistance (Fig. 3), and the Trees-NeighborsMatch dataset (Fig A4). We also outperform other rewiring methods (Drew, PR-MPNN, AMP) and graph transformers (GPS, Exphormer, GRIT, etc.) on long-range tasks (Table 3) and various molecular datasets (QM9 in Table 2, ZINC and OGB-Molhiv in Table 4, TUDataset in Table A6).  Thus, we argue that IPR-MPNNs are an efficient, state-of-the-art class of MPNNs for both common molecular datasets and those requiring long-range interactions, with strong empirical evidence that indicates an alleviation of over-squashing.
>
> 4. **The paper is not theoretical enough for NeurIPS:** NeurIPS Reviewer Guidelines [NS24-1] state that "_claims can be supported by **either theoretical analysis or experimental results**_". NeurIPS has historically included many empirical works and remains a generalist conference, suitable for applications and general machine learning topics, as per the NeurIPS 2024 Call for Papers [NS24-2].
>
> We continue addressing the raised methodology issues in the next official comment.

---

> ### Author Response · Authors · 2024-08-06
> **Rebuttal - P2 (cont. Methodology)**
>
> * **Connections with [GR24, SN24]: We want to reiterate that [GR24, SN24] were not publicly available at the time of submission.** Investigating the connections between IPR-MPNNs and these works would be very interesting. [GR24] is similar to IPR-MPNNs in its virtual node interpretation but differs fundamentally as it uses spectral layers to connect virtual nodes to the base graph, while we use a probabilistic, differentiable k-subset sampling approach to connect the base graph to the virtual graph in a data-driven manner. Our method is also similar to [SN24] since they also employ a form of heterogeneous message-passing. However, they do not use multiple virtual nodes and do not sparsify the connections between the original graph and the virtual nodes. Therefore, the theory in [SN24] is not directly applicable to our probabilistic approach.
>
> We will update the manuscript with a discussion regarding our connections with the two papers.
>
> ---
>
> * **Quadratic worst-case complexity:** We acknowledge in our Supplementary that the worst-case complexity of our model is quadratic. However, it is quadratic only in the number of virtual nodes, thereby quadratic in the number of original nodes only if the number of virtual nodes is greater or equal to the number of original nodes. **If the original graph is complete, MPNNs also have quadratic complexity in the number of nodes. Any rewiring method that might lead to a complete graph is potentially quadratic.** This is an improbable scenario for both general rewiring methods and IPR-MPNNs. We demonstrate that using up to 8 virtual nodes provides state-of-the-art performance (Table A5) while remaining sub-quadratic in the number of original nodes. We also analyze performance for up to 30 virtual nodes in Table A9, showing runtime and memory improvements over Graph Transformers even in the worst-case scenario.
>
> ---
>
> * **Virtual nodes are bottlenecks:** The primary motivation for our probabilistic layer that selectively connects the initial graph to the virtual nodes is to alleviate potential information bottlenecks between the base and virtual graphs. This acts as a sparsification operation, reducing the information transmitted to individual virtual nodes. This is different from [SN24] since they analyze a single virtual node connected to the entire original graph. Our hierarchical, heterogeneous message-passing scheme allows for bigger hidden dimensions for the virtual graph, which can also help alleviate bottlenecks. Empirically, we show that a low number of virtual nodes is sufficient for passing long-range information in most practical scenarios. For example, in the Trees-NeighborsMatch results (Figure A4), IPR-MPNNs perform as well as a Transformer up to a depth of 6, using only two virtual nodes.
>
> We thank the reviewer for the question - this is a very good point, and we will add a more nuanced discussion about possible bottlenecks in the virtual graph in our next paper revision.
>
> ---
>
> * **No comparison with [ARZ22]:** We compare with publicly available results of various rewiring approaches across multiple datasets: Drew, SPN, and PR-MPNN on QM9 (Table 2); Drew, PR-MPNN, and AMP on LRGB (Table 3); PR-MPNN on ZINC and OGB-Molhiv (Table 4); and FoSR (Table A6). Re-running all possible baselines is very expensive and time-consuming. [ARZ23] contains some results that we can compare with, and we have also performed new experiments with IPR-MPNNs on the heterophilic datasets reported in their paper.
>
> As can be seen in the next official comment, we obtain significantly better results than DiffWire on the molecular and heterophilic datasets.
>
> ---
>
> * **No comparisons with [GR24, SN24]:** The authors of the two parallel works, which became available after the NeurIPS submission deadline, also report results on peptides. As can be seen in the next official comment, our method obtains overall better scores on the Peptides datasets when compared with [SN24], and is competitive with [GR24], obtaining slightly worse results on func, and slightly better results on struct.
>
> We will update the tables for the revision, including results from [ARZ22, SN24, GR24] in our paper.
>
> ---
>
> * **No runtime comparison with other rewiring methods:** We compare with one rewiring method (PR-MPNN) in our extended runtime analysis (Table A9 in the Supplementary). Additionally, we include comparisons with the base GINE in both tables as a lower bound for required resources. In all scenarios, IPR-MPNNs' overhead over the base GINE model is generally low (see Table 1 and Table A9).
>
> We have also performed an additional experiment comparing IPR-MPNNs with Drew [GE23]. We fix the same number of message-passing layers with a similar number of parameters, and we run the Drew model from the code provided by the authors on the same hardware. The two methods are comparable, with IPR-MPNNs being slightly faster, with slightly higher memory requirements (please see the next official comment).

---

> ### Author Response · Authors · 2024-08-06
> **Rebuttal - New comparisons and experiments**
>
> ---
> **Comparison with DiffWire [ARZ22] on molecular datasets:**
>
> | Datasets | GAP (R) [ARZ22] 	| GAP (N) [ARZ22] 	| IPR-MPNN 	|
> |----------|-------------|-------------|--------------|
> | MUTAG	&uarr;| 86.9 ± 4.0 | 86.9 ± 4.0  | **98.0 ± 3.4**   |
> | PROTEINS &uarr;| 75.0 ± 3.0 | 75.3 ± 2.1 | **85.4 ± 4.4**   |
>
> ---
> **Comparison with various methods (incl. DiffWire, SDRF) on the heterophilic WebKB datasets**
>
> | Methods      | Cornell &uarr;   | Texas &uarr; | Wisconsin &uarr; |
> |-------------------------------|-------------------------|-------------| --- |
> | GINE                          	| 0.448±0.073             | 0.650±0.068 	| 0.517±0.054 |
> | DIGL [GR19] 			| 0.582±0.005             | 0.620±0.003 	| 0.495±0.003|
> | Geom-GCN [PI20]  		| 0.608±N/A                | 0.676±N/A    	| 0.641±N/A |
> | SDRF [TG21]  			| 0.546±0.004             | 0.644±0.004 	| 0.555±0.003 |
> | DiffWire [ARZ22]  			| 0.690±0.044		| N/A		| 0.791±0.021 |
> | GPS [RK22] 			| 0.718±0.024 		| 0.773±0.013  | 0.798±0.090 |
> | Graphormer [YG21] 		| 0.683±0.017		| 0.767±0.017	| 0.770±0.019|
> | IPR-MPNN 			| __0.764±0.056__ 	| __0.808±0.052__ | __0.804±0.052__ |
>
> ---
>
> **Comparison with S2GCN [GR2024] and GatedGCN+VN$_G$ [SN2024] on the long-range peptides datasets.**
>
> | Model                      	| Func  &uarr;              	| Struct &darr;            	|
> |--------------------------------|-------------------------|------------------------|
> | S2GCN [GR2024]              	| 0.7275 ± 0.0066     	| 0.2467 ± 0.0019    	|
> | S2GCN+PE [GR2024]           	| **0.7311 ± 0.0066**     	| 0.2447 ± 0.0019    	|
> | GatedGCN+PE+VN$_G$ [SN2024, best] | 0.6822 ± 0.0052     	| 0.2458 ± 0.0006    	|
> | IPR-MPNN                   	| 0.7210 ± 0.0039     	| **0.2422 ± 0.0007**    	|
>
> ---
>
> **Runtime comparison with the Drew rewiring method [GE23].**
>
> | Feature  | IPR-MPNN     	| Drew [GE23]        	|
> |----------|------------------|------------------|
> | Model	| IPR-MPNN     	| Drew         	|
> | Params   | 536k         	| 522k         	|
> | Mem  	| 1.9GB        	| 1.8GB        	|
> | s/train  | 2.98s ±0.02  	| 3.20s ±0.03  	|
> | s/val	| 0.27s ±0.00  	| 0.36s ±0.00  	|

---

> ### Author Response · Authors · 2024-08-06
> **Rebuttal - References**
>
> [SN24]: Southern, J. et al. “Understanding Virtual Nodes: Oversmoothing, Oversquashing, and Node Heterogeneity”, arXiv 2024
>
> [GR24]: Geisler, S. et al., “Spatio-Spectral Graph Neural Networks”, arXiv 2024
>
> [GE23]: Gutteridge, B. et al., “DRew: Dynamically Rewired Message Passing with Delay”, ICML 2023
>
> [BGN22]: Brüel-Gabrielsson, R. et al., “Rewiring with Positional Encodings for Graph Neural Networks”, TMLR 2023
>
> [BA17]: Battaglia, P.W. et al., “Relational inductive biases, deep learning, and graph networks”, arXiv 2018
>
> [BE23]: Banerjee, P.K. et al., “Oversquashing in GNNs through the lens of information contraction and graph expansion”, ALLERTON 2022
>
> [KR22]: Karhadkar, K. et al., “FoSR: First-order spectral rewiring for addressing over-squashing in GNNs”, ICLR 2023
>
> [ARZ22]: Arnaiz-Rodriguez, A. et al., “DiffWire: Inductive Graph Rewiring via the Lovász Bound”, LoG 2022
>
> [VR23]: Velingker, A. et al., “Affinity-Aware Graph Networks”, NeurIPS 2023
>
> [BO24]: Barbero, F. et al., “Locality-Aware Graph Rewiring in GNNs”, ICML 2024
>
> [QN24]: Qian, C. et al., “Probabilistically-Rewired Message-Passing Neural Networks”, ICLR 2024
>
> [PM17]: Pham, T. et al., “Graph Classification via Deep Learning with Virtual Nodes”, arXiv 2017
>
> [LI17]: Li, J. et al., “Learning graph-level representations for drug discovery”, arXiv 2017
>
> [CI23]: Cai, C. et al., “On the connection between MPNN and Graph Transformers”, ICML 2023
>
> [GR19]: Gasteiger, J., Weißenberger, S., and Günnemann, S. "Diffusion improves graph learning." NeurIPS 2019.
>
> [PI20]: Pei, H., et al. "Geom-gcn: Geometric graph convolutional networks." ICLR 2020.
>
> [TG21]: Topping, J., et al. "Understanding over-squashing and bottlenecks on graphs via curvature." ICLR 2022.
>
> [RK22]: Rampášek, L., et al. "Recipe for a General, Powerful, Scalable Graph Transformer." NeurIPS 2022.
>
> [YG21]: Ying, C., et al. "Do Transformers Really Perform Bad for Graph Representation?" NeurIPS 2021.
>
> [NS24-1]: NeurIPS 2024 Reviewer Guidelines
>
> [NS24-2]: NeurIPS 2024 Call For Papers

---

> > ### Comment · Reviewer_5s4E · 2024-08-12
> >
> > Thanks for your thoughtful rebuttal. It looks like some of my questions/points were valid and some had to do with a misunderstanding of the work as it is currently presented. Thank you for taking the time to explain which was which. I also acknowledge some misunderstanding of some parts of the paper on my part. I reply to specific parts where I still do not fully agree with the authors:
> >
> > **[Related work]** Regarding related work, I pointed out several (+10, most of them not concurrent and even *seminal*) works that are quite aligned and related to the topic the authors propose (rewiring, analysis of why and how rewiring works and virtual nodes). So I still think my point about missing related work was valid. Regarding the concurrent work, of course I agree and I am aware with the authors that these 2 works are concurrent. However, I think that these works, especially SN24, are of special interest since they provide theoretical insights and limitations and the interplay of virtual nodes and the over-problems. Even if this work is concurrent, the theoretical insights identified in this paper are crucial for this work. They find that a virtual node mitigates over-squashing only for some graphs. They discover that the average commute time is reduced when a virtual node is added, and also that the Jacobian between 2 nodes is independent of the distance between them if they are separated by more than 2 hops after a virtual node is added (only for some classes of graphs).
> >
> >
> > **[Theoretical explanations of VN-Overproblems]** I still maintain my position as the authors do not prove theoretically that their method solves over-quashing (as SN24 does), but they do an empirical analysis about it. Don't get me wrong, I don't think a deeper theoretical analysis is necessary for the proposed paper, but the authors claim several times that "the method reduces over-..." just based on experimental results. Considering that a theoretical analysis might be aimed for a follow-up paper, I suggest the authors to lower the claims about solving over-smoothing and over-squashing and frame it as a result of the empirical work. Good empirical work can stand on its own, especially in practically oriented papers like this one, but the statements need to be in tone with the paper and the community in which it is presented.
> >
> >
> > ***After reading the explanations of my concerns, the discussion of related work, the discussion of introducing bottlenecks, and the extensive additional experiments (runtime and baselines) performed by the authors, I raise my score from 3 to 5 and advocate acceptance of the paper.***.

---

> > > ### Author Response · Authors · 2024-08-12
> > >
> > > We thank the reviewer for their positive re-assessment of our work!
> > >
> > > As we mentioned in the rebuttal, we will include the missing related work and the concurrent works of [SN24, GR24] in the final version of the paper. We agree that there are some strong connections between our work and [SN24, GR24], and we will add a discussion where we highlight the similarities and differences between the approaches.
> > >
> > > We will also lower our claims by clarifying that we only have proof for higher expressiveness in the WL sense but no direct theoretical evidence that we alleviate OSM/OSQ. We will only highlight that the empirical evidence and performance indicate that this _might_ be the case.
> > >
> > > For further work, it would also be very interesting to investigate whether the theoretical results from [SN24] could be extended to our probabilistic method and if our sparsification via sampling strategy adds any theoretical benefits, as is the case for WL expressiveness.
> > >
> > > Once again, we thank the reviewer for their feedback and for engaging with us during the rebuttal discussions period!

---

> ### Comment · Area_Chair_T8dj · 2024-08-11
> **Please Engage in Discussion**
>
> Dear Reviewer 5s4E,
>
> Thank you for your time and efforts throughout the review period. Please read the authors' rebuttal as soon as possible and indicate if their responses have addressed all your concerns.
>
> Best,
>
> Area Chair

---

### Official Review · Reviewer_YQwu · 2024-07-03

**Soundness:** 3
**Presentation:** 2
**Contribution:** 3
**Rating:** 5
**Confidence:** 3

**Summary:**

The paper proposes implicitly rewired message-passing neural networks (IPR-MPNNs), which integrate implicit probabilistic graph rewiring into MPNNs. This method involves introducing a small number of virtual nodes into the graph, allowing for long-distance message propagation without the quadratic complexity associated with graph transformers.

**Strengths:**

1. The introduction of implicit probabilistic graph rewiring via virtual nodes is a novel contribution that addresses the limitations of traditional MPNNs, such as under-reaching and over-squashing, without incurring the high computational cost of graph transformers.
2. The authors provide a theoretical analysis demonstrating that IPR-MPNNs exceed the expressive capacity of standard MPNNs, typically limited by the 1-dimensional Weisfeiler-Leman algorithm.

**Weaknesses:**

1. It remains unclear to me how the unnormalized node priors $\theta$ are derived. On Line 170, the authors mentioned that $\theta \in \Theta$, but what is $\Theta$? According to Line 169, is it the output of the upstream GNN? Why can we trust that it accurately reflects the original node-virtual node connectivity? Is there any related optimization objective?

2. Gradient estimators for graph sampling are not a new topic. The authors may refer to [1, 2]. Providing related ablations might be more convincing.

3. I do not fully understand Figure 2. As stated by the authors, the values in the figure are about "the two most distant nodes," but why is this referred to as model sensitivity? How is this sensitivity defined?

4. Although the authors provided a complexity analysis, I believe it is necessary to provide numerical comparisons, such as the per epoch time and wall-clock time before and after using IPR-MPNN.


[1] Robust Graph Representation Learning via Neural Sparsification, NeurIPS22

[2] Differentiable Graph Module (DGM) for Graph Convolutional Networks, TPAMI

**Questions:**

1. Can you provide an algorithm table that details the training process of IPR-MPNN？

**Limitations:**

Yes

---

> ### Author Rebuttal · Authors · 2024-08-06
>
> We thank the reviewer for their feedback on our work. In the following, we will try to address the concerns raised by the reviewer:
>
> - **W1:** How are the unnormalized priors $\theta$ obtained? Is there any related optimization objective?
>
> - **RW1:** The unnormalized priors are the output of the upstream MPNN. Specifically, after some message-passing steps, we use a shared linear readout layer for each node to obtain $\theta$. If we have $N$ virtual nodes, we obtain a $1\times N$ unnormalized prior $\theta$ for each of the original nodes. These priors are then sent to the sampling layer, ensuring exactly $k$ virtual nodes are selected for each original node. There is no related optimization objective for the priors; we estimate the gradients for the sampling layer using an exactly-$k$ gradient estimator (SIMPLE).
> ---
> - **W2:** Gradient estimation for graph sampling is not a new topic.
>
> - **RW2:** We agree that gradient estimation for graph sampling is not novel by itself. We discuss similar approaches in the “Graph Structure Learning” section of Related Work in the Supplementary Materials (lines 636-648). However, our work's novelty lies in leveraging differential $k$-subset sampling for probabilistic rewiring and incorporating hierarchical message passing in an end-to-end trainable framework. We show how this probabilistic framework increases model expressiveness and benefits modeling long-range interactions. Additionally, our work differs from previous approaches by using state-of-the-art $k$-subset sampling algorithms (SIMPLE), while most previous works use $1$-subset sampling and estimate gradients with the Gumbel-Softmax trick.
>
> Regarding possible ablations - we show how the model sensitivity changes when different numbers of virtual nodes are used in Figure 2.
>
> We also provide a new ablation for the number of virtual nodes and samples. Our approach treats the virtual nodes and original nodes as heterogeneous nodes, therefore, for fair comparison, we use the same heterogeneous MPNN on MPNN+VN experiments, without reducing the number of parameters.
>
> | Model           	| ZINC  &darr;           	| ogb-molhiv  &uarr;     	| peptides-func &uarr;  	| peptides-struct &darr;	|
> |---------------------|----------------------|----------------------|---------------------|---------------------|
> | 1VN - FC        	| 0.074 ± 0.002    	| 0.753 ± 0.011    	| 0.7039 ± 0.0046 	| 0.2435 ± 0.0007 	|
> | IPR-MPNN 2VN1S  	| 0.072 ± 0.004    	| 0.762 ± 0.014    	| 0.7146 ± 0.0055 	| 0.2472 ± 0.0014 	|
> | IPR-MPNN 2VN2S  	| **0.067 ± 0.001**    	| **0.788 ± 0.006**    	| **0.7210 ± 0.0039** 	| **0.2422 ± 0.0007** 	|
>
> | Model	| peptides-func &uarr;  	|
> |----------|---------------------|
> | 1VN - FC | 0.7039 ± 0.0046 	|
> | 2VN1S	| 0.7146 ± 0.0055 	|
> | 2VN2S	| **0.7210 ± 0.0039** 	|
> | 2VN4S	| 0.7145 ± 0.0020 	|
>
> In most practical scenarios that we have tested on, having more than 2 samples doesn’t positively affect the performance but can lower the standard deviation between runs. We see that using 4 samples decreases overall performance, but the standard deviation for the 5 runs is much lower.
>
> ---
>
> - **W3:** Sensitivity analysis is unclear.
>
> - **RW3:** We detail this on lines 756-758 of our Supplementary Materials. We compute the logarithm of the symmetric sensitivity between the most distant nodes $u$, $v$, i.e., $ln\left(\left| \frac{\partial \mathbf{h}^{l}_v}{\partial \mathbf{h}^{k}_u} \right| + \left| \frac{\partial \mathbf{h}^{l}_u}{\partial \mathbf{h}^{k}_v} \right|\right)$, where $k$ to $l$ represent the intermediate layers. We thank the reviewer for pointing out that this is unclear; we will move these details to the main text of our work for the final version of our paper.
>
> ---
>
> - **W4:** Runtime comparisons for IPR-MPNN.
>
> - **RW4:** The paper includes two comparisons. The first, in Table 1, is between the base GINE model, IPR-MPNNs, and the GPS Graph Transformer. We show that our method has similar inference times and memory consumption to the base model, while being significantly more efficient than GPS. The second comparison is in Table A9 in the Supplementary Materials, where we compare various configurations of IPR-MPNN with the base GINE model, variants of the SAT Graph Transformer, GPS, and variants of the PR-MPNN rewiring method. In most cases, we perform better than the other models while maintaining similar performance and memory consumption to the base GINE model.
>
> ---
>
> - **Q1:** Algorithm table for training IPR-MPNN.
>
> - **RQ1:** We thank the reviewer for their suggestion; we will include an algorithm table/pseudocode for IPR-MPNN in the final version of our paper.
>
> ---
>
> We want to thank the reviewer for their suggestions and kindly ask the reviewer to increase their score if they are satisfied with our response. We are happy to answer any remaining questions!

---

> > ### Comment · Reviewer_YQwu · 2024-08-09
> >
> > Thank you for your response. After reading the response and other reviewers' comments, I choose to maintain my score.

---

> > > ### Author Response · Authors · 2024-08-12
> > >
> > > We once again thank the reviewer for their suggestions and questions!

---

### Official Review · Reviewer_bePC · 2024-07-04

**Soundness:** 4
**Presentation:** 4
**Contribution:** 3
**Rating:** 7
**Confidence:** 4

**Summary:**

The paper introduces a method (IPR-MPNN) which connects nodes to a small number of virtual nodes in a learnable way. The proposed approach is more expressive than MPNN whilst circumventing quadratic complexity. It can reduce oversquashing and performs well on various benchmarks.

**Strengths:**

- The benefits of the approach in the context of MPNNs (Expressivity) and Graph Transformers (Complexity) are well argued and I particularly like the probabilistic argument to ensure that isomorphic graphs are not separated with high probability.
- The results are extremely good and the approach while quite simple, achieves SOTA on various benchmarks.
- To the best of my knowledge the approach is novel due to the learnable rewiring, whilst there are interesting connections to other methods (Hierarchical MPNNs, node marking etc) which will interest the community.

**Weaknesses:**

- Whilst the method is shown well in the context of MPNNs and GTs, the probabilistic rewiring is less well motivated in the context of standard virtual nodes. Adding a baseline of MPNN + VN (with your number of VNs) would help and your results seem like they would be much better. Additionally, explaining why only connecting a fraction of nodes to the VN (instead of all) helps in terms of expressivity could also be mentioned/explored - this seems like it would be the case intuitively. You could also compare the effective resistance of your IPR-GNN graph to one where we just have a single global VN. As MPNN + VN is quite popular in the community, this would help us understand why we should favour your approach for certain tasks.
- In line 68, you suggest that your approach helps with "scaling to large graphs effectively". To me, it is not clear if this is the case. Firstly, the results seem to be mainly on small molecular datasets. Additionally, whilst you only use a small number of VNs on these datasets (eg. 2), This can still be 10% of the number of nodes in the graph. It is not clear if for large graphs you also need to have 10% VNs compared to total number of nodes. If this is the case, would your approach still be practical?

**Questions:**

- Do you have some explanation for why your approach improves over GTs? Can GTs not replicate this method?
- It would be useful to the community to mention the fraction of nodes which are connected to the VNs on some of these datasets. Connecting to a single node seems similar to node-marking and to all nodes would be equivalent to standard VN. It is interesting to see how this approach falls in this region for certain tasks.

**Limitations:**

some limitations have been addressed in the appendix. For example: "We have designed our rewiring method specifically to solve long-range graph-level tasks" - I assume the model is flexible here though in that it could also only use local interactions by connecting to. single node or node neighbours.

---

> ### Author Rebuttal · Authors · 2024-08-06
>
> We thank the reviewer for their positive assessment of our work. In the following, we will respond to the raised issues:
>
> - **W1:** Empirical and theoretical comparison with MPNN + VN.
>
> - **RW1:** We thank the reviewer for the suggestion.  The paper currently contains one such comparison in Table 2 - R-GIN-FA is a Relational GIN model with a fully-connected virtual node. Nevertheless, we also conducted the following experiments with the base model and a fully-connected virtual node, which we will include in the final version of our paper:
>
> Our approach treats the virtual nodes and original nodes as heterogeneous nodes, therefore, for fair comparison, we use the same heterogeneous MPNN on MPNN+VN experiments, without reducing the number of parameters.
>
> | Model           	| ZINC &darr;            	| ogb-molhiv &uarr;      	| peptides-func &uarr;  	| peptides-struct &darr;	|
> |---------------------|----------------------|----------------------|---------------------|---------------------|
> | 1VN - FC        	| 0.074 ± 0.002    	| 0.753 ± 0.011    	| 0.7039 ± 0.0046 	| 0.2435 ± 0.0007 	|
> | IPR-MPNN 2VN1S  	| 0.072 ± 0.004    	| 0.762 ± 0.014    	| 0.7146 ± 0.0055 	| 0.2472 ± 0.0014 	|
> | IPR-MPNN 2VN2S  	| **0.067 ± 0.001**    	| **0.788 ± 0.006**    	| **0.7210 ± 0.0039** 	| **0.2422 ± 0.0007** 	|
>
>
> The advantages of having sparse, learnable connections over fully-connected virtual nodes are as follows:
>
> * **Bottlenecks** - The network is less likely to suffer from information bottlenecks in the virtual graph since not all original nodes are connected to the virtual nodes.
>
> * **Expressiveness** - adding virtual nodes connected to the entire base graph would not make the MPNN more powerful than 1-WL. Consider the following proof sketch:
>
> Assume that we have a graph $G$ with some stable $1$-WL coloring. Now, for each node $v$ we recursively unroll the neighborhoods into a rooted tree of sufficient depth. The isomorphism type of this tree is $1$-to-$1$ to the stable color of node $v$. Now, if we consider all vertices, we have a forest of unrolling trees. We create a new vertex (the virtual node) and connect the roots of the trees in the forest to this new vertex. Then, the isomorphism type (of this forest) is $1$-to-$1$ to the isomorphism type of the original graph modulo $1$-WL; therefore, if some other graph with a virtual node has the same $1$-WL coloring, the two graphs remain indistinguishable.
>
> On the other hand, we can attach the virtual node to the original graph and unrolling from the virtual node. However, this tree will be isomorphic to the above constructed tree (forest + root vertex). Therefore, the graphs remain indistinguishable.
>
> We will add a more nuanced discussion regarding the comparison with other virtual node approaches in the final version of our paper.
>
> ---
>
> - **W2:** Would the approach be practical for large graphs?
>
> - **RW2:** We thank the reviewer for pointing out the inconsistency. We have performed new experiments on heterophilic datasets, which have a significantly greater number of nodes per graph. We report here the results:
>
> | Methods      | Cornell &uarr;   | Texas &uarr; | Wisconsin &uarr; |
> |-------------------------------|-------------------------|-------------| --- |
> | GINE                          	| 0.448±0.073             | 0.650±0.068 	| 0.517±0.054 |
> | DIGL [GR19] 			| 0.582±0.005             | 0.620±0.003 	| 0.495±0.003|
> | Geom-GCN [PI20]  		| 0.608±N/A                | 0.676±N/A    	| 0.641±N/A |
> | SDRF [TG21]  			| 0.546±0.004             | 0.644±0.004 	| 0.555±0.003 |
> | DiffWire [ARZ22]  			| 0.690±0.044		| N/A		| 0.791±0.021 |
> | GPS [RK22] 			| 0.718±0.024 		| 0.773±0.013  | 0.798±0.090 |
> | Graphormer [YG21] 		| 0.683±0.017		| 0.767±0.017	| 0.770±0.019|
> | IPR-MPNN 			| __0.764±0.056__ 	| __0.808±0.052__ | __0.804±0.052__ |
>
> Moreover, we also report results on the newly-introduced heterophilic datasets from [PV23]. These datasets contain more challenging scenarios and have a significantly higher number of nodes per graph (22K for roman-empire, 11k for tolokers, 24k for amazon-ratings, 10k for minesweeper). We can run all IPR-MPNN experiments with a memory consumption of at most 10GB.
>
> | Model        	| Roman-empire &uarr;           	| Tolokers &uarr;               	| Minesweeper &uarr;            	| Amazon-ratings &uarr;         	|
> |------------------|-----------------------------|-----------------------------|-----------------------------|-----------------------------|
> | GINE (base)  	| 0.476±0.006             	| 0.807±0.006             	| 0.799±0.002             	| **0.488±0.006**         	|
> | IPR-MPNN (ours)  | **0.839±0.006**         	| **0.820±0.008**         	| **0.887±0.006**         	| 0.480±0.007             	|
>
> ---
>
> - **Q1:** Why do we see improvements over GTs?
>
> - **RQ1:** This is a great question. We posit that the improvements come from having a stronger graph inductive bias than most GTs, where the graph is assumed to be complete. By performing all of our computations using message-passing, we keep a strong inductive bias, while the virtual nodes with learnable connections act similarly to a hard, sparse attention mechanism, thereby allowing for long-range messages to be passed while still maintaining a strong locality bias. Graph Transformers might be able to replicate this method, assuming a sparsification of the attention matrix, with a bias towards the GT “latent graph” being more similar to the original graph, thereby maintaining a better graph bias.
>
> ---
>
> - **Q2:** How many virtual nodes are used?
>
> - **RQ2:** In Table A5 from the Supplementary Materials, we report the number of virtual nodes used for most experiments. A low number (2-8 virtual nodes) is effective for most tasks.
>
> ---
>
> We want to thank the reviewer for their suggestions and kindly ask the reviewer to increase their score if they are satisfied with our response. We are happy to answer any remaining questions!

---

> > ### Comment · Reviewer_bePC · 2024-08-08
> >
> > Thank you very much for the detailed response to my questions.
> >
> > The discussion of the advantages of having sparse, learnable connections over fully-connected virtual nodes will indeed make the paper stronger and the results look good for large graphs. Adding the number of virtual nodes (as a fraction of total number of nodes) used will be useful here in understanding the scaling.
> >
> > Because of this I have raised my score and will push for acceptance.

---

> > > ### Author Response · Authors · 2024-08-08
> > >
> > > We once again thank the reviewer for their positive assessment of our work and for their comments, which helped us improve our paper!

---

> ### Author Response · Authors · 2024-08-06
> **Rebuttal - References**
>
> [GR19]: Gasteiger, J., Weißenberger, S., and Günnemann, S. "Diffusion improves graph learning." NeurIPS 2019.
>
> [PI20]: Pei, H., et al. "Geom-gcn: Geometric graph convolutional networks." ICLR 2020.
>
> [TG21]: Topping, J., et al. "Understanding over-squashing and bottlenecks on graphs via curvature." ICLR 2022.
>
> [ARZ22]: Arnaiz-Rodríguez, A., et al. "Diffwire: Inductive graph rewiring via the Lovász bound." LoG 2022.
>
> [RK22]: Rampášek, L., et al. "Recipe for a General, Powerful, Scalable Graph Transformer." NeurIPS 2022.
>
> [YG21]: Ying, C., et al. "Do Transformers Really Perform Bad for Graph Representation?" NeurIPS 2021.
>
> [PV23]: Platonov, O., et al. "A critical look at the evaluation of GNNs under heterophily: Are we really making progress?", ICLR 2023

---

### Official Review · Reviewer_UsNW · 2024-07-13

**Soundness:** 2
**Presentation:** 3
**Contribution:** 2
**Rating:** 5
**Confidence:** 4

**Summary:**

This paper introduces Implicitly Rewired Message-Passing Neural Networks (IPR-MPNNs) to address the limitations of traditional MPNNs, such as under-reaching and over-squashing. By integrating implicit probabilistic graph rewiring and adding a small number of virtual nodes, IPR-MPNNs enable long-distance message propagation without quadratic complexity. This approach enhances expressiveness and computational efficiency.

**Strengths:**

1. The paper is well-written.
2. The experimental performance is good, demonstrating promising improvements on molecule datasets.
3. The method effectively overcomes the quadratic complexity of graph transformers while capturing long-range information.

**Weaknesses:**

1. **Risk of Over-Smoothing**:
   - While IPR-MPNNs alleviate squashing and under-reaching, the approach of connecting all nodes via virtual nodes increases the risk of over-smoothing. A discussion on over-smoothing analysis would strengthen the paper.

2. **Applicability to Node-Level Tasks**:
   - The method should be directly applicable to node-level tasks, where long-range interactions are particularly beneficial. Conducting experiments on node-level datasets requiring long-range interactions and comparing with existing works would enhance the paper.

3. **Unclear Method Description**:
   - The benefit of sampling \( H \) \( q \) times, its implications, and the associated costs and disadvantages need further clarification.
   - It is unclear how to ensure that the union of the inverse assignments covers the entire set of original nodes.
   - The initial set of \( G_c \) is not clearly defined.

4. **Runtime Measurement**:
   - Better reporting of runtime measurements with respect to the number of \( q \) would provide valuable insights.

5. **Notation Confusion**:
   - There is some notation confusion, such as between \( h^{(t)} \) and \( H^{(q)} \), which needs to be clarified.

6. **Optimal Parameters and Ablation Studies**:
   - Reporting the optimal \( m \) and \( q \) for each experiment and conducting ablation studies on these parameters would provide insights on how to choose them effectively.

**Questions:**

1. Does the method exacerbate over-smoothing?
2. Why is it necessary to sample $q$ times? Does this serve a similar function as multi-head attention?
3. How do you ensure that the union of the inverse assignments covers the entire set of original nodes?
4. What is the initial set of $G_c$ ?
5. What is the runtime when $q$ is large?
6. How does the method perform on node-level tasks that require long-range interactions?
7. Can more ablation studies be conducted to provide additional insights?

**Limitations:**

yes

---

> ### Author Rebuttal · Authors · 2024-08-06
>
> We thank the reviewer for their assessment and feedback. In the following, we respond to the reviewer’s questions. Please note that the tables containing new experiments are in the next official comment and the pdf file attached to the “Global Response”.
>
> - **Q1, Q2:** Do IPR-MPNNs increase the risk of over-smoothing? Experiments on node-level tasks would strengthen the paper.
>
> - **R1, R2:** The reviewer raises a valid concern. Over-smoothing happens when nodes close to each other collapse to similar representations after several message-passing steps.
>
> We provide intuitive arguments and empirical results showing our method does not increase the risk of over-smoothing:
>
> * Over-smoothing is more likely if the virtual complete graph connects to the entire original graph since all nodes would get the same embedding from the virtual nodes during the final update. We sparsely connect original nodes to virtual nodes, which can be seen as a clustering step. This allows distant nodes of the same class to directly pass messages via the same virtual node, potentially reducing over-smoothing when compared to using virtual nodes connected to the entire graph. Therefore, it should be easier to find IPR-MPNN configurations that better handle heterophilic, long-range tasks.
>
> * To empirically address the reviewers' concern and to verify whether IPR-MPNNs can deal with node classification, we perform experiments on the WebKB datasets and some newly-proposed heterophilic datasets [PV23], which contain node classification tasks in heterophilic scenarios. We generally outperform the methods we compare with. We leave the results in the next official comment.
>
> Thank you for your question. We will update the manuscript to include a discussion on over-smoothing and the new experiments, showing improvements over baselines on heterophilic, node-level tasks.
>
> ---
>
> - **Q3:** The benefits and drawbacks of sampling are unclear.
>
> - **R3:** We thank the reviewer for raising the concern about clarity. In the following, we will try to highlight the benefits and drawbacks of sampling:
>
> * **Benefits**: the main advantage of sampling is obtaining multiple virtual node assignments for the base nodes. They can then be used to obtain different final embeddings for the same graph. From an optimization perspective, sampling multiple configurations makes it easier for the upstream model to explore the combinatorial solution space. From a representation perspective, it effectively augments the data with multiple virtual node configurations. As the reviewer pointed out, this could function similarly to multi-head attention with shared parameters.
>
> * **Costs**: we add all of the sampled configurations to the same batch, feeding the batch to the downstream model. The sample embeddings are computed in parallel, and the downstream parameters are shared. The overhead for this is typically low. The computation times for multiple samples and multiple virtual nodes are available in Table A9 in our Supplementary material. We leave the tables in the next official comment for convenience.
>
> In scenarios with 2-4 virtual nodes and 1-2 samples, our method greatly improves computation time over GraphGPS, with memory usage similar to GINE. Even with 30 virtual nodes and 20 samples, we remain faster and more memory-efficient than GraphGPS.
>
> ---
>
> - **Q3.2:** It is unclear how to ensure that the union of the inverse assignments covers the entire set of original nodes.
>
> - **R3.2:** We thank the reviewer for pointing out that this is unclear. When sampling virtual nodes, we sample exactly $k$ virtual nodes for each of the original nodes; therefore, each original node will have connections to exactly $k$ nodes in the virtual graph. We will clarify the text on lines 183-191 in the camera ready: “(...) where each original node $v\in V(G) := [n]$ is assigned to $k \in [m]$ virtual nodes, _therefore, all of the nodes in original graph will have exactly $k$ edges connecting to the virtual graph._”
>
> ---
>
> - **Q3.3:** The C(G) set of virtual nodes is not well defined.
>
> - **R3.3:** Thank you for pointing to this unclarity. The virtual node set C(G) is a set of new nodes that form a complete virtual graph - we will expand the text on lines 183-191 for the next revision: “and a _fixed, new_ virtual node set $C(G):=[m]$ of cardinality $m$”
>
> ---
>
> - **Q4:** Runtimes with respect to the number of samples.
>
> - **R4:** We report runtimes based on the number of samples and virtual nodes in Table A9 of our Supplementary materials. We will move some results from the Appendix to the main paper for the final revision.
>
> ---
>
> - **Q5:** $H^{(q)}$ and $h^{(t)}$ are a confusing notation.
>
> - **R5:** Thank you for highlighting the unclear notation. $H^{(q)}$ represents the matrix of new connections between original and virtual nodes, while $h^{(t)}$ represents the hidden embeddings of the original nodes after $t$ message-passing layers. We will clarify this in the final version.
>
> ---
>
> - **Q6:** Optimal hyperparameters and ablation studies.
>
> - **R6:** The hyperparameters used in our experiments are in Table A5 of the Supplementary materials. We also conducted new experiments with a virtual node connected to the entire virtual graph and with two virtual nodes, sampling one, two, or four configurations. Please see the next official comments for the results.
>
> In most practical scenarios that we have tested on, having more than 2 samples doesn’t positively affect the performance but can lower the standard deviation between runs. We see that using 4 samples decreases overall performance, but the standard deviation for the 5 runs is much lower.
>
> ---
>
> Once again, we thank the reviewer for their suggestions, and we kindly ask the reviewer to increase their score if they are satisfied with our response. We are happy to answer any remaining questions!

---

> > ### Comment · Reviewer_UsNW · 2024-08-13
> >
> > I appreciate the author’s detailed response to my questions and comments, as well as the additional experimental results. However, I still have concerns about oversmoothing. Incorporating a measurement based on Dirichlet Energy might provide further insights. After considering the feedback from other reviewers, I have decided to maintain my current score.

---

> ### Author Response · Authors · 2024-08-06
> **Rebuttal - New experimental results**
>
> **Q1, Q2: New experimental results on the WebKB heterophilic datasets and on the heterophilic datasets proposed in [PV23]:**
>
>
> | Methods      | Cornell &uarr;   | Texas &uarr; | Wisconsin &uarr; |
> |-------------------------------|-------------------------|-------------| --- |
> | GINE                          	| 0.448±0.073             | 0.650±0.068 	| 0.517±0.054 |
> | DIGL [GR19] 			| 0.582±0.005             | 0.620±0.003 	| 0.495±0.003|
> | Geom-GCN [PI20]  		| 0.608±N/A                | 0.676±N/A    	| 0.641±N/A |
> | SDRF [TG21]  			| 0.546±0.004             | 0.644±0.004 	| 0.555±0.003 |
> | DiffWire [ARZ22]  			| 0.690±0.044		| N/A		| 0.791±0.021 |
> | GPS [RK22] 			| 0.718±0.024 		| 0.773±0.013  | 0.798±0.090 |
> | Graphormer [YG21] 		| 0.683±0.017		| 0.767±0.017	| 0.770±0.019|
> | IPR-MPNN 			| __0.764±0.056__ 	| __0.808±0.052__ | __0.804±0.052__ |
>
>
> | Model        	| Roman-empire &uarr;           	| Tolokers &uarr;               	| Minesweeper  &uarr;           	| Amazon-ratings &uarr;         	|
> |------------------|-----------------------------|-----------------------------|-----------------------------|-----------------------------|
> | GINE (base)  	| 0.476±0.006             	| 0.807±0.006             	| 0.799±0.002             	| **0.488±0.006**         	|
> | IPR-MPNN (ours)  | **0.839±0.006**         	| **0.820±0.008**         	| **0.887±0.006**         	| 0.480±0.007             	|
>
> ---
>
> **Q4: Reduced runtime report, also available in Table A9 of the Appendix:**
>
>
>
> | Model  	| #Params | V. Nodes | Samples | Train s/ep      	| Val s/ep        	| Mem. usage |
> |------------|---------|----------|---------|---------------------|---------------------|------------|
> | GINE   	| 502k	| -    	| -   	| 3.19 ± 0.03     	| 0.20 ± 0.01     	| 0.5GB  	|
> | GraphGPS   | 558k	| -    	| -   	| 17.02 ± 0.70    	| 0.65 ± 0.06     	| 6.6GB  	|
> | IPR-MPNN   | 548k	| 2    	| 1   	| 7.31 ± 0.08     	| 0.35 ± 0.01     	| 0.8GB  	|
> | IPR-MPNN   | 548k	| 4    	| 2   	| 7.37 ± 0.08     	| 0.35 ± 0.01     	| 0.8GB  	|
> | IPR-MPNN   | 549k	| 30   	| 20  	| 9.41 ± 0.38     	| 0.43 ± 0.01     	| 1.2GB  	|
>
>
> ---
>
> **Q6: Table comparing a virtual node connected to the entire graph (1VN - FC) with IPR-MPNN with two virtual nodes, one sample (2VN1S), two samples (2VN2S) and four samples (2VN4S)**
>
> | Model           	| ZINC &darr;            	| ogb-molhiv &uarr;      	| peptides-func &uarr;  	| peptides-struct 	&darr;|
> |---------------------|----------------------|----------------------|---------------------|---------------------|
> | 1VN - FC        	| 0.074 ± 0.002    	| 0.753 ± 0.011    	| 0.7039 ± 0.0046 	| 0.2435 ± 0.0007 	|
> | IPR-MPNN 2VN1S  	| 0.072 ± 0.004    	| 0.762 ± 0.014    	| 0.7146 ± 0.0055 	| 0.2472 ± 0.0014 	|
> | IPR-MPNN 2VN2S  	| **0.067 ± 0.001**    	| **0.788 ± 0.006**    	| **0.7210 ± 0.0039** 	| **0.2422 ± 0.0007** 	|
>
>
> | Model	| peptides-func &uarr;  	|
> |----------|---------------------|
> | 1VN - FC | 0.7039 ± 0.0046 	|
> | 2VN1S	| 0.7146 ± 0.0055 	|
> | 2VN2S	| **0.7210 ± 0.0039** 	|
> | 2VN4S	| 0.7145 ± 0.0020 	|

---

> ### Author Response · Authors · 2024-08-06
> **Rebuttal - References**
>
> [GR19]: Gasteiger, J., Weißenberger, S., and Günnemann, S. "Diffusion improves graph learning." NeurIPS 2019.
>
> [PI20]: Pei, H., et al. "Geom-gcn: Geometric graph convolutional networks." ICLR 2020.
>
> [TG21]: Topping, J., et al. "Understanding over-squashing and bottlenecks on graphs via curvature." ICLR 2022.
>
> [ARZ22]: Arnaiz-Rodríguez, A., et al. "Diffwire: Inductive graph rewiring via the Lovász bound." LoG 2022.
>
> [RK22]: Rampášek, L., et al. "Recipe for a General, Powerful, Scalable Graph Transformer." NeurIPS 2022.
>
> [YG21]: Ying, C., et al. "Do Transformers Really Perform Bad for Graph Representation?" NeurIPS 2021.
>
> [PV23]: Platonov, O., et al. "A critical look at the evaluation of GNNs under heterophily: Are we really making progress?", ICLR 2023

---

> ### Comment · Area_Chair_T8dj · 2024-08-11
> **Please Engage in Discussion**
>
> Dear Reviewer UsNW,
>
> Thank you for your time and efforts throughout the review period. Please read the authors' rebuttal as soon as possible and indicate if their responses have addressed all your concerns.
>
> Best,
>
> Area Chair

---

> ### Author Response · Authors · 2024-08-13
>
> We thank the reviewer for their suggestion.
>
> While our method is not tailored to alleviate over-smoothing, we agree that having evidence that our model does not _increase_ the phenomena of over-smoothing is beneficial.
>
> We have measured the Dirichlet Energy of the final layer on the heterophilic WebKB Cornell, Wisconsin, Texas datasets and on the roman-empire dataset from [PV23]. We compare IPR-MPNN with the base GINE model.
>
> The results are averaged for 5 runs, and we also report the standard deviations:
>
> | Model       | Cornell ↑             | Wisconsin ↑           | Texas ↑               | Roman-Empire ↑        |
> |-------------|-----------------------|-----------------------|-----------------------|-----------------------|
> | GINE (base) | 8.34 ± 4.74           | 5.09 ± 3.11           | 7.41 ± 4.57          | 19.36 ± 1.16          |
> | IPR-MPNN    | **12.10 ± 4.77**          | **8.89 ± 2.67**           | **9.65 ± 1.90**           | **49.49 ± 2.41**          |
>
> As can be observed, IPR-MPNNs have higher Dirichlet Energy on Cornell, Wisconsin, Texas and significantly higher on roman-empire, indicating that over-smoothing is alleviated for these datasets.
>
> Please note that IPR-MPNN also obtains much better overall results on these datasets (see previous comment).
>
> We will add these results to the final version of the paper. Overall, the empirical evidence indicates that IPR-MPNNs _do not_ increase over-smoothing, but they rather slightly alleviate the phenomena.
>
> Thank you for helping us improve our work. We kindly ask the reviewer to reconsider their assessment and increase their score if they find our response convincing. We are open to answering any remaining questions if time permits.

---

> > ### Comment · Reviewer_UsNW · 2024-08-13
> >
> > Thank you to the authors for addressing my concern with the oversmoothing measurement. It has alleviated my concerns. I will maintain my score.

---

### Author Rebuttal · Authors · 2024-08-06

We thank the reviewers for their thoughtful comments and suggestions. We believe that our paper is much stronger after considering reviewer feedback.

All of the new comparisons and experiments are included in the one-page pdf attached to the global response.

---

In the following, we summarize the modifications that will be included in our final paper:

**UsNW, bePC, 5s4E:** New experiments on heterophilic datasets (WebKB, [PV23]) - Tables 10, 11 in the pdf. We show that we are able to run our model on large graphs and that, in most cases, we obtain better results on heterophilic datasets when compared to baselines, indicating that IPR-MPNNs might alleviate over-smoothing.

**UsNW, bePC, YQwu:** New experiments where we compare with a simple MPNN + VN and we change the number of samples that we use in Tables 13,15. We observe that, generally, having only two samples achieves very good performance with low computation overhead. We will also add a discussion regarding how we are different from MPNN+VN and about the benefits and costs of sampling to the final version of the paper.

**UsNW:** We will fix the clarity issues on lines 183-191 and the confusing notation.

**YQwu:** We will clarify how we compute the sensitivity in the final version of our paper.

**YQwu:** We will add an algorithm table/pseudocode for IPR-MPNN in the final version of our paper.

**5s4E:** We will add the three missing related works [ARZ22, VR23, BO24] and expand the discussions on [BE23, KR22]. We will also have a discussion regarding our connection to [GR24, SN24] - two concurrent works that were not available at submission time.

**5s4E:** We will add a more nuanced discussion about possible bottlenecks in the virtual graph.

**5s4E:** We added new comparisons with [ARZ22] in Tables 10 and 12, and with the concurrent works of [GR24, SN24] in Table 16. We show that we outperform [ARZ22, SN24] and are generally competitive with [GR24].

**5s4E:** We added a runtime and memory comparison with Drew [GE23] in Table 14. We are slightly faster than Drew but with slightly higher memory consumption.

---

We would like to once again thank the reviewers for their feedback and for helping us significantly strengthen our paper. We are happy to respond to any additional questions during the discussion period.

We kindly ask the reviewers to increase their scores if they are satisfied with our response.

---
References:

[PV23]: Platonov, O., et al. "A critical look at the evaluation of GNNs under heterophily: Are we really making progress?", ICLR 2023

[ARZ22]: Arnaiz-Rodríguez, A., et al. "Diffwire: Inductive graph rewiring via the Lovász bound." LoG 2022.

[VR23]: Velingker, A. et al., “Affinity-Aware Graph Networks”, NeurIPS 2023

[BO24]: Barbero, F. et al., “Locality-Aware Graph Rewiring in GNNs”, ICML 2024

[BE23]: Banerjee, P.K. et al., “Oversquashing in GNNs through the lens of information contraction and graph expansion”, ALLERTON 2022

[KR22]: Karhadkar, K. et al., “FoSR: First-order spectral rewiring for addressing over-squashing in GNNs”, ICLR 2023

[GE23]: Gutteridge, B. et al., “DRew: Dynamically Rewired Message Passing with Delay”, ICML 2023

[GR24]: Geisler, S. et al., “Spatio-Spectral Graph Neural Networks”, arXiv 2024

[SN24]: Southern, J. et al. “Understanding Virtual Nodes: Oversmoothing, Oversquashing, and Node Heterogeneity”, arXiv 2024

---

### Decision · Program_Chairs · 2024-09-25

**Decision:**

Accept (poster)

**Comment:**

This paper proposed implicitly rewired message-passing neural networks (IPR-MPNNs), which integrate implicit probabilistic graph rewiring into MPNNs to alleviate the under-reaching and over-squashing issues. Most of the reviewers agreed that the proposed approach is novel and this paper is technically solid. All the reviews tend to accept the paper during the reviewer-author discussion phase. Therefore, I recommend this paper to the NeurIPS 2024 conference.